# OmniActor: A Generalist GUI and Embodied Agent for 2D&3D Worlds

**Longrong Yang**[1], **Zhixiong Zeng**[2,*], **Yufeng Zhong**[2], **Jing Huang**[2], **Liming Zheng**[2], **Lei Chen**[2], **Haibo Qiu**[2], **Zequn Qin**[3,†], **Lin Ma**[2,†], **Xi Li**[1,†]

[1]College of Computer Science and Technology, Zhejiang University
[2]Meituan Technology
[3]School of Software Technology, Zhejiang University

## Abstract

Multimodal large language models are progressively advancing toward *multimodal agents* that can proactively execute tasks. Existing research on multimodal agents primarily targets either GUI or embodied scenarios, corresponding to interactions within 2D virtual world and 3D physical world, respectively. However, many real-world tasks inherently require agents to interleave interactions across both types of environments. We initially mix GUI and embodied data to train models, but find performance degradation caused by data conflicts. Further analysis reveals that GUI and embodied data exhibit *synergy* at shallow layers but *conflict* at deep layers, resembling the cerebrum-cerebellum mechanism in the human brain. To this end, we introduce a high-performance generalist agent, **OmniActor**, designed from both structural and data perspectives. First, we propose Layer-heterogeneous MoE that separates parameters at deep layers to eliminate conflict, while sharing parameters at shallow layers to leverage synergy. This design enables OmniActor to outperform agents trained solely on GUI or embodied data in their respective tasks. Furthermore, we unify the action spaces of GUI and embodied tasks and collect large-scale datasets from diverse sources for training. This substantially enhances the performance of OmniActor across various scenarios, especially in GUI tasks.

## 1 Introduction

Foundation models have demonstrated remarkable capabilities in both language and vision understanding tasks. In particular, Multimodal Large Language Models (MLLMs) (Liu et al., 2023b; 2024; Wang et al., 2024; Bai et al., 2025; Team et al., 2025), *i.e.*, multimodal foundation models trained on massive image-text corpora, have achieved strong performance in image-text understanding benchmarks. As an extension of MLLMs, multimodal agents have been successfully applied to graph user interface (GUI) control (Wu et al., 2025b; Xu et al., 2025; Qin et al., 2025) and embodied intelligence (Shah et al., 2023; Ha et al., 2023; Kim et al., 2025), and are increasingly adopted in industrial applications.

Humans naturally perform actions across diverse environments, such as shopping in a 2D digital world or interacting with entities in a 3D physical world. This raises a natural question: *can an agent act in both 2D and 3D environments like humans, thereby realizing a generalist agent?* This goal is non-trivial due to the inherent differences between 2D and 3D worlds. GUI and embodied tasks serve as typical tasks for evaluating agent abilities in 2D and 3D worlds, respectively. Several preliminary attempts have been made to unify GUI and embodied actions within a single agent. For instance, Magma (Yang et al., 2025a) represents GUI and embodied actions as set-of-mark and trace-of-mark, and uses videos to enrich action supervision. GEA (Szot et al., 2025) introduces a continuous multi-embodiment tokenizer for mixed action prediction, combined with an online reinforcement learning phase to further improve performance.

---

[†]Corresponding authors. [*]Projector leader.

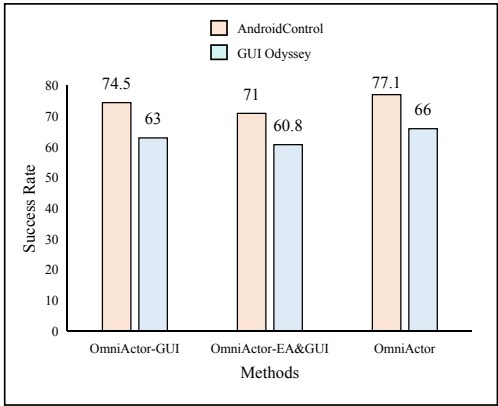 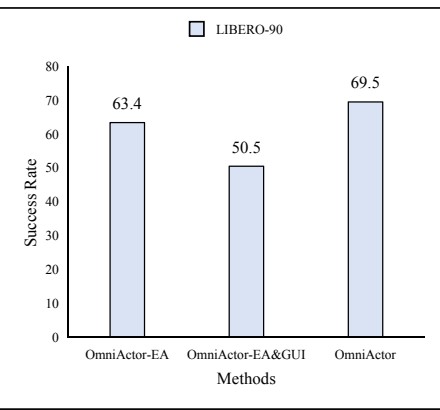

(a) GUI tasks in 2D digital world        (b) Embodied tasks in 3D physical world

Figure 1: Performance comparison. AndroidControl and GUI Odyssey are evaluation benchmarks for GUI tasks. LIBERO-90 is a evaluation benchmark for embodied tasks. OmniActor-GUI denotes the agent trained solely on GUI data. OmniActor-EA denotes the agent trained solely on embodied data. OmniActor-EA&GUI is trained jointly on both data sources. OmniActor, equipped with the proposed Layer-heterogeneity MoE, effectively leverages synergy while mitigating conflict, leading to substantial improvements.

When unifying data from distinct environments into one model, a fundamental issue is *conflict* and *synergy* across these data. Our initial experiments reveal that naively mixing GUI and embodied data leads to degraded performance due to conflict, as shown in Figure 1. We attribute the conflict to the significant action differences: GUI actions are typically text-described operations (*e.g.*, "click"), whereas embodied actions involve continuous 6-DoF end-effector displacements. On the other hand, GUI and embodied tasks should also exhibit synergy, as they share a similar task structure (environments, instructions, and actions). The comprehension of environments and instructions can mutually reinforce each other across embodied and GUI tasks. Unfortunately, existing generalist agents lack explicit mechanisms to manage such conflict and synergy, resulting in suboptimal performance.

To this end, we propose a novel generalist agent, termed **OmniActor**. We not only propose Layer-heterogeneity MoE to leverage the synergy between GUI and embodied data while mitigating their conflict, but also collect and unify large-scale data to train a high-performance unified agent. Specifically, $(i)$ **Layer-heterogeneity MoE**. By examining the parameter update directions across different data, we observe that the directions from GUI and embodied data are more consistent in shallow layers than in deep layers. This phenomenon is analogous to the cerebrum-cerebellum mechanism in humans: the cerebrum, being closer to the sensory input, undertakes a comprehensive understanding of environments and instructions, whereas the cerebellum, being closer to the motor output, is responsible for executing diverse actions. Thus, we share parameters in shallow layers to exploit the synergy between GUI and embodied data, while separating parameters in deeper layers to avoid conflicts caused by divergent action requirements. As shown in Figure 1, this design substantially improves the agent performance. $(ii)$ **Large-scale GUI and embodied data**. We curate large-scale training data from multiple sources, including GUI data from OS-Atlas Wu et al. (2025b), Uground Gou et al. (2025), Aguvis Xu et al. (2025), and Aria-UI Yang et al. (2025b), as well as embodied data from LIBERO Liu et al. (2023a). All data are unified into a standardized format, where each sample contains a system prompt, an image, a task instruction, and an output action. We employ a text tokenizer for GUI actions and a specialized embodied tokenizer for embodied actions, both mapped into a shared vocabulary. By unifying the data format and action space, we enable training on large-scale GUI and embodied data, thereby enhancing the agent across diverse tasks.

In summary, our contributions are as follows:

- We propose the Layer-heterogeneity MoE, which shares parameters in the shallow layers to exploit the synergy between GUI and embodied data, while separating parameters in the deep layers to mitigate the conflict arising from action discrepancies.

- We expand and unify GUI and embodied data by standardizing data formats and action spaces. This enables training with large-scale datasets from both domains, substantially enhancing the performance of agents on GUI and embodied tasks.

- We extend the capabilities of MLLMs to build a generalist agent, OmniActor. OmniActor significantly outperforms existing generalist agents in both GUI and embodied tasks, and even surpasses state-of-the-art GUI or embodied agents.

## 2 RELATED WORKS

### 2.1 GUI AGENT

Early GUI agents (Deng et al., 2023; Gur et al., 2024; Lai et al., 2024; He et al., 2024; Yang et al., 2023) rely on HTML or AXTree data to describe GUI elements through textual descriptions. These methods depended on the structured representation of web page elements, enabling agents to locate targets based on tags, attributes, or text content. However, these methods need meticulous pre-processing designs, limiting their generalization under different scenarios. With the rise of multimodal large language models (MLLMs), an increasing number of pure vision-based methods have been proposed. Leveraging the strong visual capabilities of MLLM, these methods eliminate the need for manually designing data pre-processing schemes for each scenario, and thus have significant advantages over HTML-dependent GUI agents in terms of generalization. Early pure visual GUI agents (Hong et al., 2024; Zhang & Zhang, 2024; Zhang et al., 2024; Wu et al., 2025a; 2024; 2025a) focus on using MLLMs to process GUI screenshots to understand GUI components, replacing HTML-dependent GUI component understanding. Furthermore, researchers find that scaling data is crucial for enhancing GUI agents, thus proposing the use of synthetic data or video data (Yang et al., 2025b; Xie et al., 2025; Wu et al., 2025b; Sun et al., 2025). For instance, Aria-UI (Yang et al., 2025b) introduces an extensible data synthesis pipeline for generating grounding instruction samples, which are used to train MLLMs specifically for GUI grounding. OS-Atlas (Wu et al., 2025b) releases the first multi-platform GUI data synthesis toolkit, supporting the automatic synthesis of cross-platform GUI grounding data while resolving action naming conflicts.

Recently, researchers have aimed to make GUI agents more similar to human-like agents (Gou et al., 2025; Xu et al., 2025; Lu et al., 2025; Qin et al., 2025). For example, UGround (Gou et al., 2025) proposes to execute actions only using human-like keyboard and mouse operations, demonstrating the feasibility of human-like GUI agents. Aguvis (Xu et al., 2025) unifies different GUI action spaces and divides training into grounding and planning, enhancing the planning capabilities of agents after they have acquired strong GUI grounding abilities. GUI-Odyssey (Lu et al., 2025) proposes a large-scale cross-application training and evaluation dataset, allowing agents to interleavely use multiple applications to perform tasks instead of being limited to a single application. UI-TARS (Qin et al., 2025) can perform human-like interactions, which achieves accurate perception of GUI elements by collecting a large amount of GUI screenshots and enhances agent capabilities through multiple reasoning modes. UItron Zeng et al. (2025) achieves the state-of-the-art performance through system-level design and engineering optimization. MobileDreamer Cao et al. (2026) enhances long-horizon performance for mobile GUI agents by integrating an efficient textual-sketch world model and rollout imagination to enable future-aware decision-making. GUIRoboTron-Speech Han et al. (2025b;a) empowers agents with speech recognition capabilities.

### 2.2 EMBODIED AGENT

In the field of robotics, benefiting from the rapid development of MLLMs, there has emerged a new trend of leveraging the strong generalization ability of MLLMs to enable them to act as "brains" for controlling robots to perform tasks. ATM (Wen et al., 2023) uses pre-trained models to predict trajectories in videos, providing action guidance for agents. MUTEX (Shah et al., 2023) learns strategies from multiple modalities, which can understand instructions in six modalities or their combinations. ACT (Zhao et al., 2023) proposes a low-cost system that enables robots to complete multiple high-difficulty tasks through Supervised Fine-Tuning (SFT) and a Transformer-based action chunking algorithm. Distill-D (Ha et al., 2023) efficiently generates embodied data with language labels using Large Language Models (LLMs) and sampling planners. MaIL (Jia et al., 2024) proposes a new SFT architecture based on Mamba, which reduces overfitting and enhance generalization.

Prise (Zheng et al., 2024a) regards temporal action abstraction as a sequence compression problem, and combines continuous action quantization with byte pair encoding to learn action abstractions. MDT (Reuss et al., 2024) addresses data annotation issues through potential goal-conditional state representations and can handle long-term manipulation tasks with sparsely annotated data. Then, RoboFlamingo (Li et al., 2024b), Octo (Team et al., 2024), and OpenVLA (Kim et al., 2025) are vision-language-action models built on open-source MLLMs, which can be applied to robot control through fine-tuning. GR-2 (Cheang et al., 2024) pre-trains agents on massive internet video data and fine-tune agents on embodied trajectory data. Some recent works Yan et al. (2025); Zhong et al. (2025); Zheng et al. (2025b) focus on unifying data from different sources or using cheap data to improve the performance of embodied agent.

## 2.3 GENERALIST AGENT

Researchers have recently begun to study generalist agents, aiming to develop a foundational model that serves multimodal agent tasks in both digital and physical worlds, capable of accomplishing various agent tasks ranging from GUI navigation to robotic manipulation. Magma (Yang et al., 2025a) unifies GUI and embodied tasks into State of Motion (SoM) and Trajectory of Motion (ToM), and augments action trajectory data with video data. GEA (Szot et al., 2025) utilizes a Continuous Multi-Embodiment Tokenizer to perform hybrid action prediction for embodied intelligence, games, GUI control, planning, and navigation, and sets Supervised Fine-Tuning and Online Reinforcement Learning. NaviMaster (Luo et al., 2025) holds that both GUI and embodied tasks can be formulated as Markov decision processes, thus enabling the generation of trajectories for GUI and embodied tasks in a generalist form. Based on this, it employs reinforcement learning and a distance-aware reward to enhance the generalization of the agent.

## 3 METHODOLOGY

### 3.1 OVERVIEW

**Generalist agent**: A multimodal generalist agent should be capable of interacting with both 2D virtual world and 3D real world. We define a multimodal agent $\pi$, whose inputs include the visual observation $I$ of the environment, task description $T$, and historical information $H$ in text form. Its output is a set of actions $A$, defined as:

$$A = \pi(I, T, H) = \{a_1^l, \cdots, a_N^l\}, \tag{1}$$

where $l \in \{\text{gui}, \text{robot}\}$ indicates whether the $n$-th action token $a_n$ is a GUI token or an embodied token. This formulation is applicable to different types of tasks:

- Embodied tasks in 3D real world: For instance, the embodied task $T$ may be "put the black bowl at the back on the plate". The output includes a 6-degree-of-freedom (6-DoF) displacement $(\text{pos}_x, \text{pos}_y, \text{pos}_z, \text{rot}_x, \text{rot}_y, \text{rot}_z)$ of the end-effector, and an additional dimension indicating whether the gripper is open or closed. $(\text{pos}_x, \text{pos}_y, \text{pos}_z)$ denote the spatial position of the robotic arm, and $(\text{rot}_x, \text{rot}_y, \text{rot}_z)$ denote its rotation angles. Each dimension is typically normalized to the range [-1, 1].
- GUI tasks in 2D virtual world: For example, the GUI task $T$ may be "book a hotel". The output includes both language tokens representing the semantic type of the action (e.g., "tap" or "click") and the position $(x, y)$ where the action is applied.

Given an observation image, task instruction, and historical information, a multimodal generalist agent should predict either GUI actions or embodied actions depending on the environment.

**Our method**: To train a generalist agent, **OmniActor**, that leverages the synergy between GUI and embodied tasks while mitigating conflicts, we introduce improvements in both data and structure: ($i$) Data. As shown in Figure 2, we unify GUI and embodied data into a consistent format where each sample includes a system prompt, an image, a task instruction, and an action description. Embodied actions, normalized to [-1, 1], are discretized into $K$ intervals, each corresponding to a token ID, thereby converting embodied actions into tokens. ($ii$) Structure. We propose layer-heterogeneity MoE. As illustrated in Figure 3, shallow layers are shared to exploit the synergy between GUI and

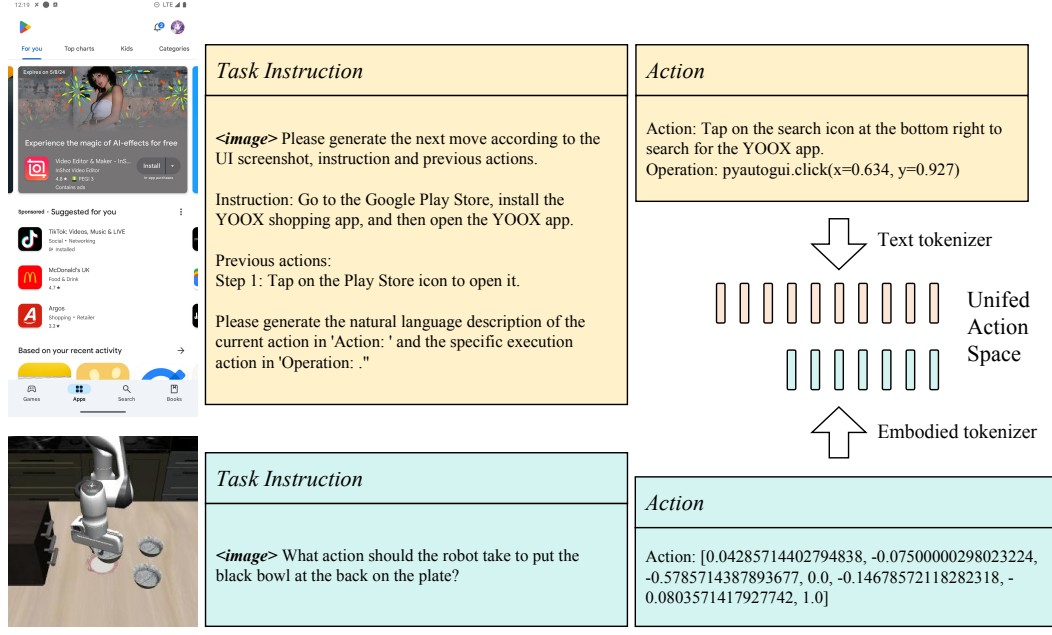

Figure 2: Data format. We unify GUI tasks in 2D digital world and embodied tasks in 3D real-world environments Each sample includes a system prompt, an image , a task instruction, and an action description. GUI actions are typically textual, and we use the MLLM tokenizer as a GUI tokenizer to convert these actions into tokens. Embodied actions are usually described by 6-DoF displacement of the end effector plus a gripper control signal, which are tokenized using an embodied tokenizer. Both GUI and embodied actions share the same vocabulary, forming a unified action space for prediction.

embodied data, while deep layers are separated to avoid conflicts arising from action differences. This design is analogous to the cerebrum-cerebellum mechanism: shallow layers (like the cerebrum) process information comprehensively, whereas deep layers (like the cerebellum) generate task-specific actions. Further details of these modules are provided in the subsequent sections.

## 3.2 DATA PROCESSING

**Unification of Data Format**: We adopt Qwen2-VL (Wang et al., 2024) and Qwen2.5-VL (Bai et al., 2025) as our base MLLMs. To ensure compatibility with Qwen2-VL and Qwen2.5-VL in LlamaFactory (Zheng et al., 2024b), we convert all GUI and embodied data into the ShareGPT format, as illustrated in Figure 2. Each sample consists of a system prompt, an image, a task instruction, and an action. The system prompt specifies the action space of the agent, while the image represents the current environment. Images are tokenized via the image encoder of the base MLLM, whereas system prompts, task instructions, and GUI actions are tokenized using the text tokenizer of the base MLLM. Embodied actions, however, require a dedicated tokenization process.

**Tokenization of embodied actions**: We represent embodied actions as discrete tokens for processing in this paper. An embodied action is defined as a 6-DoF displacement of the end effector, $(\mathrm{pos}_x, \mathrm{pos}_y, \mathrm{pos}_z, \mathrm{rot}_x, \mathrm{rot}_y, \mathrm{rot}_z)$, together with one additional dimension indicating whether the gripper is open or closed. All dimensions are normalized to lie within [-1, 1]. We uniformly discretize this interval into $K$ bins and assign token IDs to each bin by selecting the $K$ least frequent token IDs from the vocabulary of the base MLLM. For instance, as shown in Figure 2, an embodied action [0.043, -0.075, -0.579, 0.0, -0.147, -0.080, 1.0] is converted into the token sequence [151510, 151500, 151482, 151515, 151515, 151516, 151642].

Through these two steps, we standardize and tokenize both GUI and embodied data, enabling effective training of the base MLLM to build a unified generalist agent.

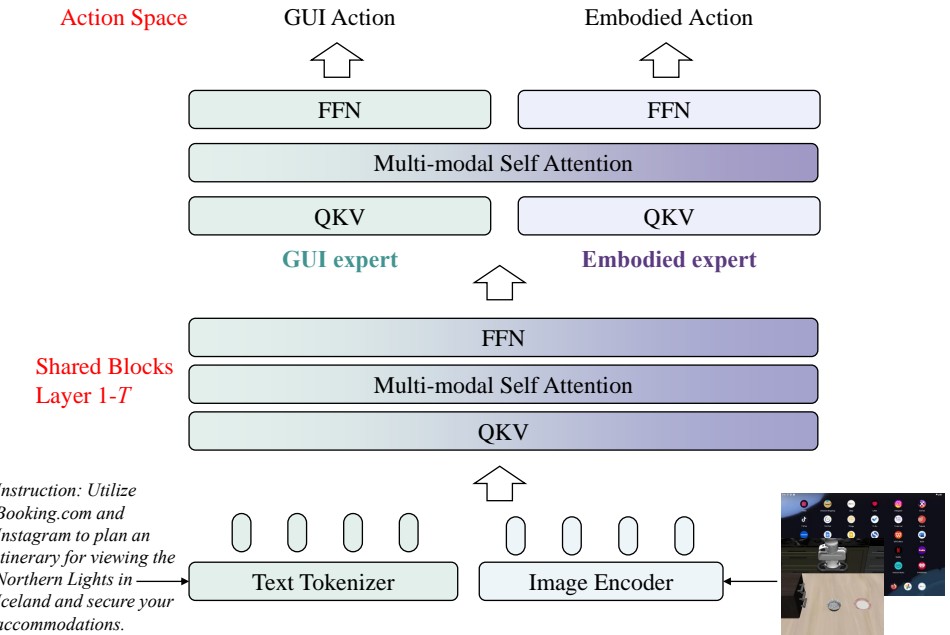

Figure 3: The proposed pipeline. The proposed **Layer-heterogeneity MoE** shares parameters in shallow layers to leverage the synergy between GUI and embodied data, and separates parameters in deep layers to eliminate conflicts between GUI and embodied data caused by action differences.

## 3.3 LAYER-HETEROGENEITY MOE

GUI and embodied data exhibit both synergy and conflict, which, from the perspective of model optimization, are reflected in the directions of parameter updates. Specifically, "synergy" refers to similar parameter update directions arising from GUI and embodied data, whereas "conflict" refers to divergent update directions between the two. From this viewpoint, if GUI and embodied data are synergistic for certain parameters, these parameters should be shared; otherwise, parameters should be separated. Motivated by this principle, we analyze the parameter update directions between GUI and embodied data and observe that the similarity of optimization directions is much higher in shallow layers compared to deep layers. We hypothesize that this phenomenon occurs because the understanding of environments and instructions is synergistic across GUI and embodied tasks, while the differences between GUI actions and embodied actions necessitate separate parameters. Thus, we propose the layer-heterogeneity MoE, designed to leverage the synergy between GUI and embodied data while mitigating their conflict, as illustrated in Figure 3.

Specifically, assuming the layer count of the LLM is $L$, we set a layer depth threshold $T$ to distinguish shared layers and separated layers. ($i$) For shallow layers $l_s$ ($1 \leq l_s \leq T$), as shown in Figure 3, we share parameters to process both GUI and embodied data, for utilizing the synergy between GUI and embodied data:

$$\mathbf{x}'_\ell = \mathrm{MSA}(\mathrm{LN}(\mathbf{x}_{\ell-1})) + \mathbf{x}_{\ell-1}, \ell \in \{1, \ldots, T\}, \tag{2}$$

$$\mathbf{x}_\ell = \mathrm{FFN}(\mathrm{LN}(\mathbf{x}'_\ell)) + \mathbf{x}'_\ell, \ell \in \{1, \ldots, T\}, \tag{3}$$

where $\mathbf{x}_\ell$ indicates the hidden representation in the layer $\ell$, MSA indicates the Multi-head Self-Attention, FFN indicates the Feed-Forward Network, and LN indicates the Layer Normalization.

($ii$) For deep layers $l_d$ ($T < l_d \leq L$), as shown in Figure 3, we separate the attention and FFN. One set of parameters (*e.g.*, $\mathrm{FFN}_{gui}$) is used to process GUI data, and another set of parameters (*e.g.*, $\mathrm{FFN}_{rob}$) is used to process embodied data:

$$\mathbf{x}'_\ell = \begin{cases} \mathrm{MSA}_{\mathrm{gui}}\left(\mathrm{LN}_{\mathrm{gui}}(\mathbf{x}_{\ell-1})\right) + \mathbf{x}_{\ell-1}, & \text{if } \mathbf{x}_{\ell-1} \text{ is GUI data} \\ \mathrm{MSA}_{\mathrm{rob}}\left(\mathrm{LN}_{\mathrm{rob}}(\mathbf{x}_{\ell-1})\right) + \mathbf{x}_{\ell-1}, & \text{if } \mathbf{x}_{\ell-1} \text{ is embodied data} \end{cases} \quad \ell \in \{T+1, \ldots, L\} \tag{4}$$

$$\mathbf{x}_\ell = \begin{cases} \text{FFN}_{\text{gui}}\left(\text{LN}_{\text{gui}}(\mathbf{x}'_\ell)\right) + \mathbf{x}'_\ell, & \text{if } \mathbf{x}'_\ell \text{ is GUI data} \\ \text{FFN}_{\text{rob}}\left(\text{LN}_{\text{rob}}(\mathbf{x}'_\ell)\right) + \mathbf{x}'_\ell, & \text{if } \mathbf{x}'_\ell \text{ is embodied data} \end{cases} \qquad \ell \in \{T+1, \ldots, L\} \qquad (5)$$

where $\text{MSA}_{gui}$ ($\text{FFN}_{gui}$, $\text{LN}_{gui}$) and $\text{MSA}_{rob}$ ($\text{FFN}_{rob}$, $\text{LN}_{rob}$) denote expert parameters for GUI and embodied tasks, respectively. Moreover, as illustrated in Figure 3, distinct prediction heads $h_{gui}$ and $h_{rob}$ are used for GUI and embodied actions. Although both action types are expressed within a unified vocabulary, their action spaces differ substantially; thus, separating heads is crucial to avoid conflicts. During inference, we assume the task type of the input sample (GUI or embodied) is known, and the corresponding branch is selected to process the data.

Through the layer-heterogeneity MoE design with shallow sharing and deep separation, we harness the synergy between GUI and embodied data while eliminating conflict, thereby enabling a high-performance generalist agent capable of operating across both 2D and 3D worlds.

## 4 EXPERIMENTS

### 4.1 EXPERIMENTAL SETUP

**Training data**: Due to the diversity and complexity of GUI components, we firstly train the base MLLM on GUI grounding data (a combination of open-source datasets OS-Atlas (Wu et al., 2025b), Uground (Gou et al., 2025), Aguvis (Xu et al., 2025), and Aria-UI (Yang et al., 2025b), similar to ScaleTrack (Huang et al., 2025)). This stage is not applied to embodied tasks, as the components in embodied scenarios are relatively simple. After the grounding stage, we train the model using trajectory data to enable it to perform tasks as an agent. For GUI tasks, we follow ScaleTrack (Huang et al., 2025) and select the Aguvis dataset (Xu et al., 2025), whereas for embodied tasks, we utilize the popular LIBERO dataset (Liu et al., 2023a). Our GUI and robot trajectory data contain 706K and 669K samples respectively, with a ratio close to 1:1. During training, embodied actions consist of a series of continuous points, which are substantially different from typical MLLM outputs. Thus, it is necessary to resample the embodied data to ensure sufficient learning of embodied tasks (Zawalski et al., 2025; Szot et al., 2025). We resample the embodied data 5 times, resulting in an approximate GUI-to-embodied data ratio of 1:5. The total data size is around 3.4M samples for grounding training and 4.1M samples for trajectory training.

**Evaluation benchmarks**: We select two datasets as the GUI evaluation benchmark: ($i$) Android Control (Li et al., 2024a). Android Control is designed to evaluate GUI agents on mobile platforms, which consists of AndroidControl-Low and AndroidControl-High. ($ii$) GUI-Odyssey (Lu et al., 2025) provides a comprehensive dataset for evaluating cross-application GUI agents. Then, we select LIBERO (Liu et al., 2023a) as the embodied evaluation benchmark, which includes a total of 90 evaluation tasks covering three different scenarios: kitchen, living room, and study room. More details please refer to the supplementary material.

**Comparison**: We conduct a comprehensive comparison with GUI, embodied, and generalist agents. ($i$) In-house GUI agents, including Claude 3.5 (Hu et al., 2024), GPT-4o (Hurst et al., 2024), and UI-TARS-7B (Qin et al., 2025). Open-sourced GUI agents, including SeeClick (Cheng et al., 2024), Aria-UI (Yang et al., 2025b), OS-Atlas-7B (Wu et al., 2025b), Aguvis-7B (Xu et al., 2025), and ScaleTrack-7B (Huang et al., 2025).

($ii$) Embodied agents, mainly for robotic arms, including MaIL (Jia et al., 2024), PRISE (Zheng et al., 2024a), MUTEX (Shah et al., 2023), ACT (Zhao et al., 2023), Distill-D (Ha et al., 2023), MDT (Reuss et al., 2024), OpenVLA (Kim et al., 2024), $\pi_0$ (Black et al., 2024), SpatialVLA (Qu et al., 2025), and UniACT (Zheng et al., 2025a).

($iii$) Generalist agents, including Magma (Yang et al., 2025a), GEA (Szot et al., 2025), and Navi-Master (Luo et al., 2025). Magma (Yang et al., 2025a) does not report results on LIBERO, Android-Control, and GUI Odyssey, but provides codes and the pre-trained model. Thus, we finetune the pre-trained model on LIBERO, AndroidControl, and GUI Odyssey to report the performance. GEA and NaviMaster do not provide codes and pre-trained models.

**Our method**: ($i$) OmniActor-GUI: Use GUI data to train Qwen2-VL (Wang et al., 2024). ($ii$) OmniActor-EA: Use embodied data to train Qwen2-VL (Wang et al., 2024). ($iii$) OmniActor: Use GUI and embodied data to train Qwen2-VL (Wang et al., 2024). Parameters in shallow layers are

Table 1: **Comparison between different agents on embodied (robot control) tasks and GUI navigation tasks.** We report the success rate on each task. "-" indicates that the agent is unable to perform the task or its performance on the task has not been reported.

| Agents | Source | Embodied Tasks | GUI Tasks | | |
| | | LIBERO-90 | AndroidControl-Low | AndroidControl-High | GUI Odyssey |
| --- | --- | --- | --- | --- | --- |
| Claude | In-house | - | 19.4 | 12.5 | 3.1 |
| GPT-4o | In-house | - | 19.4 | 20.8 | 3.3 |
| UI-TARS-7B | In-house | - | 90.8 | 72.5 | 87.0 |
| SeeClick | ACL'24 | - | 75.0 | 59.1 | 53.9 |
| Aria-UI | ACL'25 | - | 67.3 | 10.2 | 36.5 |
| OS-Atlas-7B | ICLR'25 | - | 85.2 | 71.2 | 62.0 |
| Aguvis-7B | ICML'25 | - | 80.5 | 61.5 | 63.8 |
| ScaleTrack-7B | arXiv'25 | - | 86.6 | 77.9 | 65.3 |
| MUTEX | CoRL'23 | 53.0 | - | - | - |
| Distill-D | CoRL'23 | 49.9 | - | - | - |
| ACT | RSS'23 | 46.6 | - | - | - |
| MaIL | CoRL'24 | 60.3 | - | - | - |
| PRISE | ICML'24 | 54.4 | - | - | - |
| MDT | RSS'24 | 67.2 | - | - | - |
| OpenVLA | CoRL'24 | 73.5 | - | - | - |
| $\pi_0$ | arxiv'24 | 87.3 | - | - | - |
| SpatialVLA | RSS'25 | 46.2 | - | - | - |
| UniACT | CVPR'25 | 73.0 | - | - | - |
| Magma | CVPR'25 | 34.7 | 52.1 | 32.7 | 51.0 |
| GEA | CVPR'25 | - | - | 57.3 | - |
| NaviMaster | arXiv'25 | - | 68.9 | 54.0 | - |
| OmniActor-GUI | | - | 88.4 | 74.5 | 63.0 |
| OmniActor-EA | | 63.4 | - | - | - |
| OmniActor | | - 69.5 | 87.5 | 77.1 | 66.0 |

shared. Parameters in deep layers are separated as the MoE, where GUI data is processed by one set of experts, and embodied data is processed by another set of experts.

## 4.2 MAIN RESULTS

As shown in Table 1, we have several findings: $(i)$ Existing generalist agents usually have a relatively low performance than GUI agents on GUI tasks and embodied agents on embodied tasks, although they can finish the two tasks simultaneously. $(ii)$ OmniActor-GUI achieves the comparable performance with existing GUI agents on GUI tasks. OmniActor-EA achieves the comparable performance with existing embodied agents on embodied tasks. $(iii)$ On embodied tasks, OmniActor achieves 6.1% higher task success rate than OmniActor-EA. On GUI tasks, OmniActor achieves the average 1.3% higher task success rate than OmniActor-GUI. Especially, OmniActor has a significantly higher task success rate on AndroidControl-High and GUI Odyssey, indicating the superiority on long-chain trajectory prediction.

## 4.3 STUDY

**Study about parameter sharing and separation**: To leverage the synergy between GUI and embodied data while mitigating their conflicts, we investigate strategies for parameter sharing and separation. We conduct the following experiments: $(i)$ OmniActor-EA&GUI: Directly mix GUI and embodied data to train Qwen2-VL (Wang et al., 2024). $(ii)$ OmniActor *hard*: Building on $(i)$, we modify the model to a MoE structure, where GUI data is processed by one set of experts and embodied data by another. The parameters of attention heads, FFNs, and the classification head are fully separated. $(iii)$ OmniActor: Different from $(ii)$, parameters in shallow layers are shared, and parameters in deep layers are separated. $(iv)$ OmniActor *router*: Building on $(iii)$, a router is incorporated prior to each MoE layer, which takes token features as input and outputs the probabilities

Table 2: **Comparison between different parameter sharing and separation strategies.** We report the success rate on each task.

| Models | Embodied Tasks | GUI Tasks | | | Avg |
| | LIBERO-90 | AndroidControl-Low | AndroidControl-High | GUI Odyssey | |
| --- | --- | --- | --- | --- | --- |
| OmniActor-GUI | - | 88.4 | 74.5 | 63.0 | - |
| OmniActor-EA | 63.4 | - | - | - | - |
| OmniActor-EA&GUI | 50.5 | 86.3 | 71.0 | 60.8 | 67.2 |
| OmniActor *hard* | 59.5 | 85.6 | 75.2 | 63.9 | 71.1 |
| OmniActor | 69.5 | 87.5 | 77.1 | 66.0 | 75.0 |
| OmniActor *router* | 64.0 | 86.0 | 72.6 | 66.1 | 72.2 |

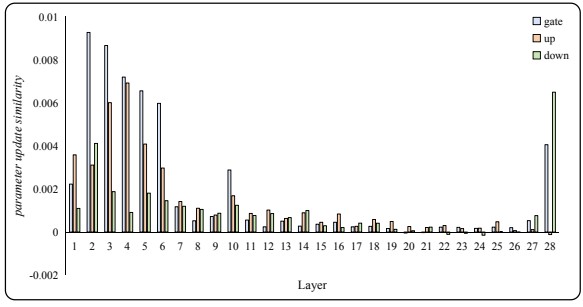

(a) *Parameter update similarity* statistic for FFN

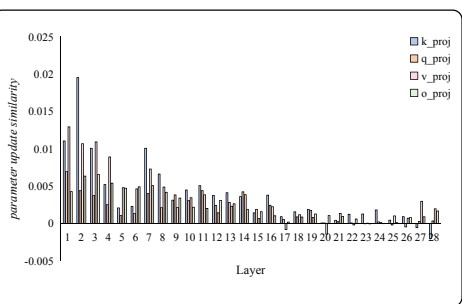

(b) *Parameter update similarity* statistic for attention

Figure 4: Statistic on *parameter update similarity*. Shallow layers exhibit significantly higher *parameter update similarities* than deep layers, motivating our design of the Layer-heterogeneity MoE.

of assigning the token to different experts. The expert with the highest probability is selected as the active expert for the token.

As shown in Table 2, we have some important findings: ($i$) Directly mixing GUI and embodied data degrades model performance due to conflicts between the two tasks. ($ii$) OmniActor *hard* separates almost all parameters for GUI and embodied tasks. Compared to mixed training, it improves performance by eliminating conflicts, but cannot exploit the synergy between GUI and embodied data. ($iii$) OmniActor outperforms separate training by leveraging the synergy between GUI and embodied data while simultaneously resolving conflict. ($iv$) OmniActor outperforms OmniActor *router*, which verifies that improvements stem from task-specific partitioning rather than MoE capacity.

**Statistic about parameter optimization directions**: We analyze the parameter optimization directions for GUI and embodied tasks to answer two key questions: *Which parameters can be shared and which should be separated?* Parameters can be shared when the update directions from GUI and embodied data are consistent; otherwise, they should be separated.

*How to identify shareable parameters?* We propose a novel metric, *parameter update similarity*, which estimates the consistency between GUI and embodied tasks for each parameter. For instance, for the gate projection in the first FFN, we compute the parameter difference before and after training with GUI data, denoted as $d_{gui}$, and similarly compute the difference after training with embodied data, denoted as $d_{robot}$. The cosine similarity between $d_{gui}$ and $d_{robot}$ defines the *parameter update similarity* for the gate proj. A high similarity indicates that the parameter can be shared; otherwise, it should be separated. When the *parameter update similarity* is high, the parameter should be shared; otherwise, the parameter should be separated. We conduct this analysis for FFNs (gate proj, up proj, and down proj) across layers in Figure 4(a) and for attention (k proj, q proj, v proj, and o proj) in Figure 4(b). Based on these statistics, shallow-layer parameters are shared, while deep-layer parameters are separated. Accordingly, we empirically set the layer depth threshold $T = 8$, which distinguishes shared from separated layers.

**Generalization verification**: We analyze the impact of training agents on different base MLLMs. Specifically, we replace the base MLLM with Qwen2.5-VL 7B (Bai et al., 2025), adjust the training

Table 3: **Study about different base MLLMs.** We evaluate different base MLLMs to verify the generalization of the proposed strategy. We report the success rate on each task.

| Base MLLMs | Embodied Tasks | GUI Tasks | | | Avg |
|---|---|---|---|---|---|
| | LIBERO-90 | AndroidControl-Low | AndroidControl-High | GUI Odyssey | |
| Qwen2-VL 7B | 69.5 | 87.5 | 77.1 | 66.0 | 75.0 |
| Qwen2.5-VL 7B | 65.2 | 87.8 | 83.2 | 81.0 | 79.3 |

data to match its input format, and migrate the Layer-heterogeneity MoE to this model. Experimental results are summarized in Table 3. When training the generalist agent on the more powerful Qwen2.5-VL 7B, the average success rate increases by 4.3% compared to training on Qwen2-VL 7B, and the average performance is close to the state-of-the-art in-house GUI agent UI-TARS-7B for GUI tasks. This verifies the generalization of OmniActor across different base MLLMs.

## 5 CONCLUSION

Our research aims to construct a high-performance generalist agent from both data and structural perspectives. Data scaling is particularly crucial for agent performance; therefore, we unify the action spaces of GUI and embodied tasks, and collect a large amount of GUI and embodied data for training the agent. Furthermore, we identify the conflicts and synergies between GUI and embodied tasks. By separating deep-layer parameters and sharing shallow-layer parameters, we leverage the synergies between tasks and eliminate conflicts. Extensive experiments demonstrate the effectiveness of the generalist agent OmniActor across different scenarios. Especially in GUI scenarios, OmniActor even outperforms many of the latest state-of-the-art GUI agents.

**Acknowledgements.** This work was supported in part by the National Science Foundation for Distinguished Young Scholars under Grant 62225605, Zhejiang Provincial Natural Science Foundation of China under Grant LD24F020016, Ningbo Science and Technology Special Projects under Grant No. 2025Z028, and the Fundamental Research Funds for the Central Universities.

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

# 6 SUPPLEMENTARY MATERIAL

In our supplementary material, we provide the following details and experiments:

- 6.1: We provide more details about hyper-parameters and evaluation benchmarks.
- 6.2: We provide more details about prompts and representative samples.
- 6.3: We provide experimental results on real-world robots.
- 6.4: We provide the study about the layer depth threshold $T$, statistical mean of performance, and different disentangling ways.
- 6.5: We discuss the in-group *parameter update similarity* and the *parameter update similarity* on the head.
- 6.6: We provide a case study.
- 6.7: We provide a visualization study about synergy and conflict phenomenon.
- 6.8: We provide details about the use of Large Language Models for this paper.

## 6.1 DETAILS ABOUT HYPER-PARAMETERS AND EVALUATION

**Training hyper-parameters**: We use the AdamW optimizer with a learning rate of 1e-5 and a cosine learning rate scheduler with a warm-up ratio of 0.03. We set the global batch size to 128 in the grounding training phase and 64 in the trajectory training phase, and adopt the DeepSpeed ZERO3-style parallel strategy. To align with GUI data, we resize all $256 \times 256$ images from LIBERO to $448 \times 448$. The embodied tokenizer uniformly discretize [-1, 1] into $K$ intervals, where $K$=256. The layer depth threshold $T$ to distinguish shared and separated layers is set to 8.

**Evaluation benchmarks**: We select two datasets as the GUI evaluation benchmark: ($i$) Android Control (Li et al., 2024a). Android Control is designed to evaluate GUI agents on mobile platforms. It consists of two types of sub-tasks: AndroidControl-Low requires the agent to predict specific action types and parameters (*e.g.*, coordinates) at each step based on an image, a task instruction, and a human-annotated natural language description of actions. AndroidControl-High requires the agent to independently plan and sequentially predict action types and parameters when only given images and task instructions. ($ii$) GUI-Odyssey (Lu et al., 2025) provides a comprehensive dataset for evaluating cross-application GUI agents. Similar to AndroidControl-High, it only provides images and task instructions, and it requires the agent to perform complex cross-application operations. Following previous studies, we randomly sample 800-step actions to create a subset for evaluation.

We select LIBERO (Liu et al., 2023a) as the embodied evaluation benchmark, which includes a total of 90 evaluation tasks covering three different scenarios: kitchen, living room, and study room. For each task, we test 20 unseen start and goal configurations. In each test, the agent first receives the task instruction and the initial scene (image), then outputs an action. The action and the scene are fed into the end effector to generates a new scene (image). The task instruction and the new scene are then fed into the agent for predicting the next action. When the task is successfully executed, the end effector returns a success signal, and the evaluation proceeds to the next task. The maximum steps allowed for task execution is 400.

## 6.2 DETAILS ABOUT SAMPLES

In this subsection, we detail the prompts we use in this work and present representative GUI and embodied samples for further clarification.

**Prompts**: We use different prompts for GUI and embodied tasks. Specifically, the system prompt for GUI tasks is "You are a GUI agent. You are given a task and a screenshot of the screen. You need to perform a series of pyautogui actions to complete the task.", and that for embodied tasks is "You are an agent that can see, talk and act. You are given a task and an image. You need to perform a series of actions to complete the task.".

The instruction format for GUI tasks is "Please generate the next move according to the UI screenshot, instruction and previous actions. \n\n Instruction: {overall_goal} \n\n Previous actions: {previous_actions} \n\n Please generate the natural language description of the cur-

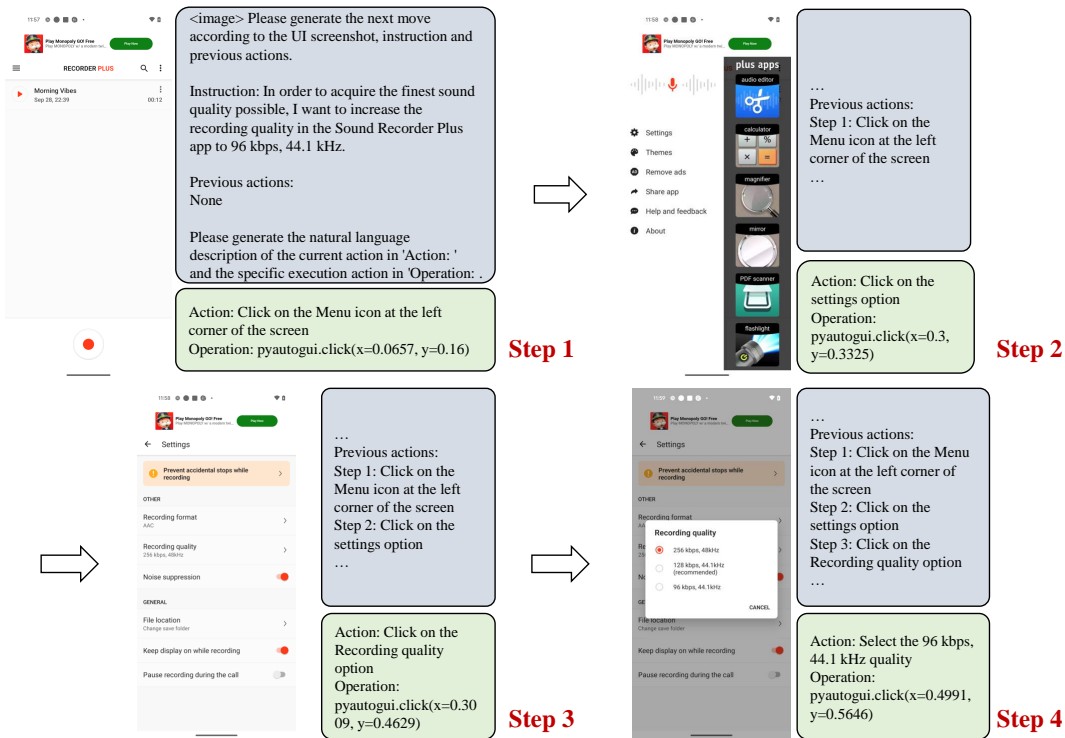

Figure 5: A representative GUI sample. Each GUI sample consists of several steps. Each step includes an image (environment observation), instruction, history (previous actions), and action.

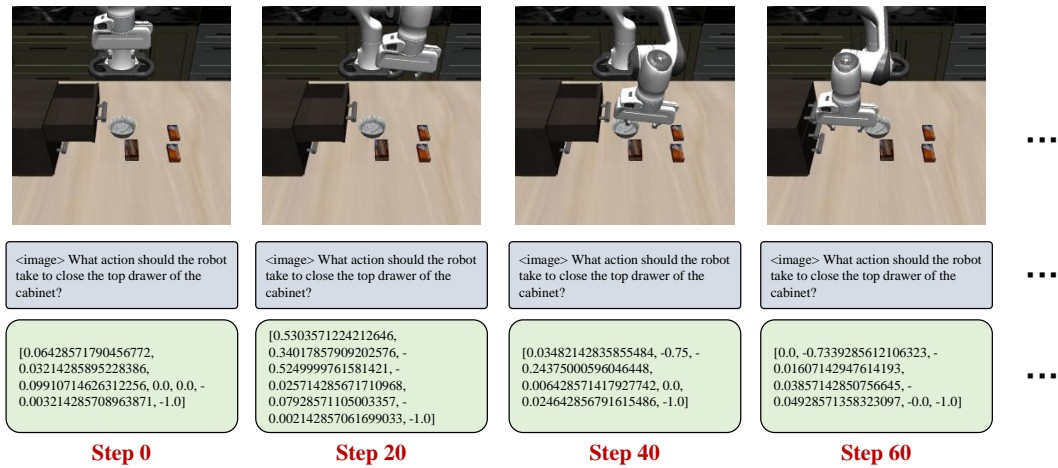

Figure 6: A representative embodied sample. Each embodied sample consists of several steps, with each step including an image (environment observation), instruction, and action.

rent action in 'Action: ' and the specific execution action in 'Operation: '.". For embodied tasks, the instruction format is "What action should the robot take to {overall_goal}?".

Each sample has different {overall_goal} and {previous_actions}. As shown in Figure 5, for the GUI sample, the {overall_goal} is "In order to acquire the finest sound quality possible, I want to increase the recording quality in the Sound Recorder Plus app to 96 kbps, 44.1 kHz.", and the {previous_actions} serves as the agent's history and grows as the number of steps increases. As shown in Figure 6, for the embodied sample, the {overall_goal} is "close

Instruction: *What action should the robot take to place rxbar blueberry into top drawer?*

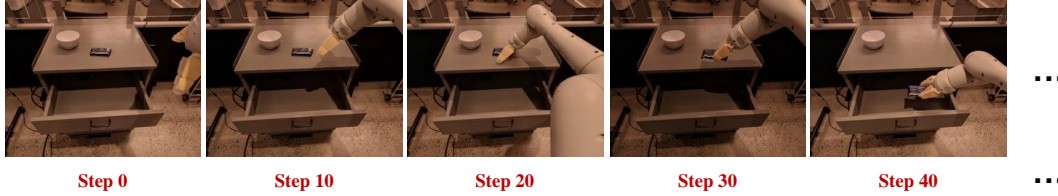

Figure 7: An example about real-world robot data.

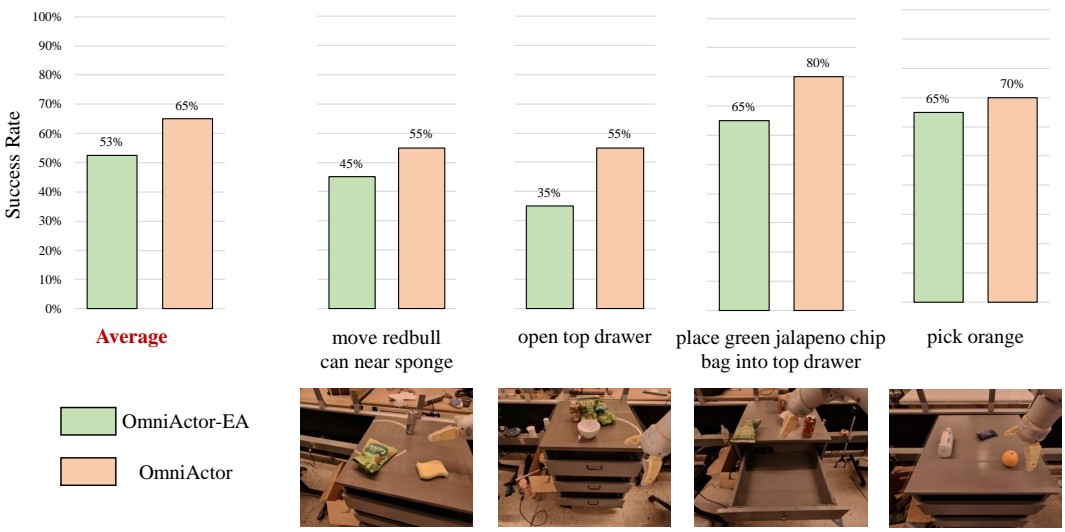

Figure 8: The model performance on real-world robot data. We evaluate OmniActor-EA and OmniActor on four tasks: "move redbull can near sponge", "open top drawer", "place green jalapeno chip bag into top drawer", "pick orange".

the top drawer of the cabinet". We maintain consistency in the system prompts and instruction formats during both training and inference.

In addition, we conduct experiments based on LlamaFactory (Zheng et al., 2024b) and largely follow the practices in LlamaFactory (Zheng et al., 2024b).

**Representative GUI and Embodied Samples**: As shown in Figure 5, each GUI sample consists of several steps. Each step includes an image (environment observation), instruction, history (previous actions), and action. Similarly, as shown in Figure 6, each embodied sample consists of several steps, with each step including an image (environment observation), instruction, and action. During the evaluation of GUI tasks, the model predicts an action given an image, instruction, and history; for embodied task evaluation, the model predicts an action given an image and instruction.

## 6.3 REAL-WORLD ROBOT EXPERIMENTS

In this subsection, we validate the generalization ability of OmniActor in 3D real-world scenarios through real-world robot experiments. Specifically, to preserve the structure of the training data, we collect approximately 700K real-world robotic trajectory data from RT-1 (Brohan et al., 2022), which is comparable in scale to the LIBERO trajectory data. Then, we convert the format of these data into the embodied data format described in the main text (as shown in Figure 6), which is used to replace the LIBERO data for training both OmniActor-EA and OmniActor. Subsequently, we evaluate the performance of the trained models on four unseen tasks: "move redbull can near sponge", "open top drawer", "place green jalapeno chip bag into top drawer", "pick orange". We perform 20 trials per task. As shown in Figure 7, we present an example of the real-world robot

Table 4: **Study about the layer depth threshold $T$.** We report the success rate on each task.

| | Embodied Tasks | GUI Tasks | | | Avg |
|---|---|---|---|---|---|
| | LIBERO-90 | AndroidControl-Low | AndroidControl-High | GUI Odyssey | |
| $T$=4 | 63.0 | 84.5 | 74.1 | 65.4 | 71.8 |
| $T$=8 | 69.5 | 87.5 | 77.1 | 66.0 | 75.0 |
| $T$=12 | 63.0 | 87.7 | 75.8 | 67.7 | 73.6 |
| $T$=20 | 61.5 | 87.3 | 71.1 | 67.6 | 72.0 |

Table 5: **Study about the statistical mean and standard deviation of the model performance.** We report the success rate on each task.

| | AndroidControl-Low | AndroidControl-High | GUI Odyssey |
|---|---|---|---|
| OmniActor-GUI | $88.82 \pm 1.27$ | $75.20 \pm 0.78$ | $63.77 \pm 0.32$ |
| OmniActor | $88.98 \pm 1.53$ | $77.30 \pm 0.85$ | $65.67 \pm 0.31$ |

Table 6: **Study about different disentangling ways.** We report the success rate on each task.

| | Embodied Tasks | GUI Tasks | | | Avg |
|---|---|---|---|---|---|
| | LIBERO-90 | AndroidControl-Low | AndroidControl-High | GUI Odyssey | |
| OmniActor | 69.5 | 87.5 | 77.1 | 66.0 | 75.0 |
| OmniActor-attention | 68.0 | 86.5 | 76.7 | 63.7 | 73.7 |
| OmniActor-FFN | 66.0 | 84.6 | 74.0 | 63.2 | 72.0 |
| OmniActor-head | 61.0 | 86.0 | 75.7 | 62.6 | 71.3 |

data, illustrating the motion of the robotic arm in a 3D real-world environment. Figure 8 compares the performance of OmniActor-EA and OmniActor, along with environmental examples of the four evaluation tasks. The experimental results demonstrate that OmniActor achieves a success rate exceeding 60%, and its performance outperforms OmniActor-EA, thereby validating the effectiveness of the Layer-heterogeneous MoE under real-world robot data.

## 6.4 FURTHER STUDY

**Layer depth threshold $T$**: The layer depth threshold $T$ is a tunable hyper-parameter. We discuss different selections of $T$ in Table 4 and find that the performance is optimal when $T$=8.

**Statistical mean of performance**: We conduct five independent training runs for OmniActor-GUI and OmniActor respectively, and calculate their statistical mean and standard deviation. As shown in Table 5, the performance gain is statistically reliable.

**Different disentangling ways**: We further explore different ways of disentangling the GUI and embodied knowledge in the Table 6. OmniActor separate the parameters of attention, FFNs, and prediction heads. OmniActor-attention shares attention while seperates the remaining parameters. OmniActor-FFN shares FFNs and OmniActor-head shares the prediction head. We find that sharing attention shows the least performance degradation, while sharing FFNs orheads leads to significant performance drops. This may be attributed to that FFNs account for a large proportion of the total parameters, and heads are crucial for prediction.

## 6.5 PARAMETER UPDATE SIMILARITY

**In-group parameter update similarity**: We have randomly divided the GUI training data into two in-groups, named GUI-1 and GUI-2 respectively. GUI-1 and GUI-2 are roughly equal in size, each accounting for 50% of the total GUI training data. Then, we train the model using GUI-1 and GUI-2 separately. Similarly, the parameter changes induced by GUI-1 and GUI-2 are denoted as $d_{gui,1}$ and $d_{gui,2}$ respectively. Then, we calculate the cosine similarity between $d_{gui,1}$ and $d_{gui,2}$ as the *parameter update similarity*. As shown in Figure 9, we find that the *parameter update similarity* across different layers is consistently high, and there is no trend of "higher similarity in shallow

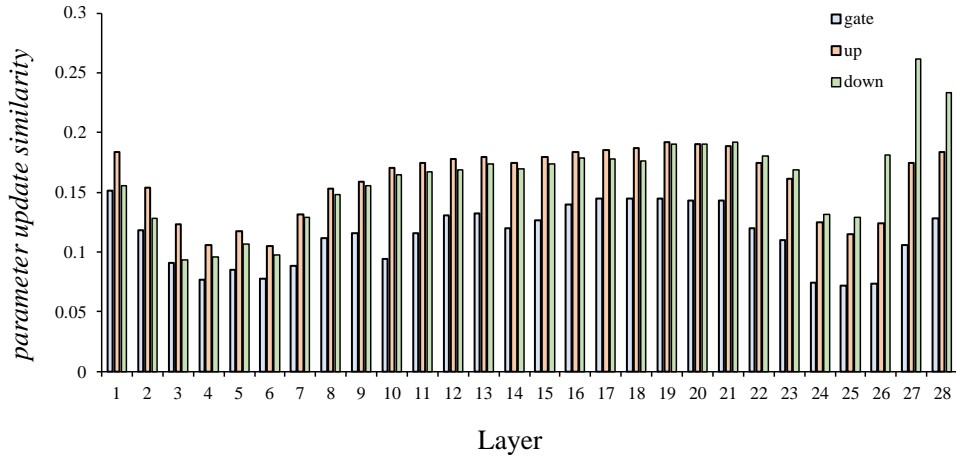

Figure 9: In-group *parameter update similarity* for FFN. There is no trend of "higher similarity in shallow layers and lower similarity in deep layers".

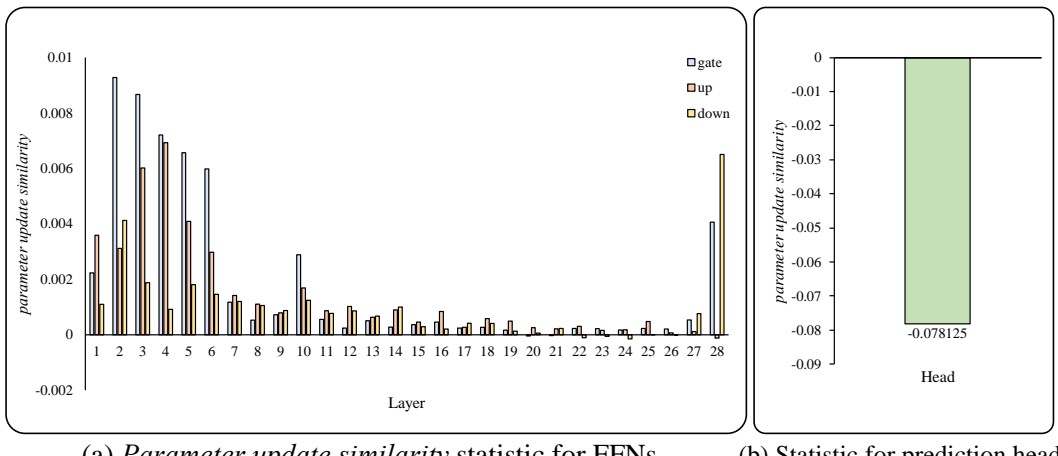

(a) *Parameter update similarity* statistic for FFNs  (b) Statistic for prediction head

Figure 10: *Parameter update similarity* for FFNs and the prediction head. Due to the action space difference, the parameter update similarity on the prediction head is significantly low.

layers and lower similarity in deep layers". This further verifies the necessity of designing layer-heterogeneity MoE for GUI and embodied data.

**Parameter update similarity on the head**: The main text of our paper does not present the *parameter update similarity* of the prediction head, and this section supplements this data point. The calculation process of the *parameter update similarity* on the prediction head is consistent with that on other parameters. As shown in Figure 10, the parameter update similarity on the prediction head is significantly low, indicating a severe conflict between GUI and embodied data on the prediction head, which results from the significant action space difference of the two types of data.

## 6.6 CASE STUDY

We further analyze conflicts and synergies between GUI and embodied tasks. As shown in Figure 11, Figure 12, and Figure 13, We provide the related case study.

**The transfer from GUI to embodied tasks**: GUI data enhance the model ability to understand key terms such as "top" in the instructions of embodied tasks, so helping the model distinguish similar objects at different positions. We present two tasks in Figure 11: one task is "close the top drawer

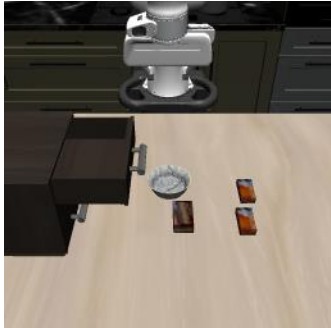 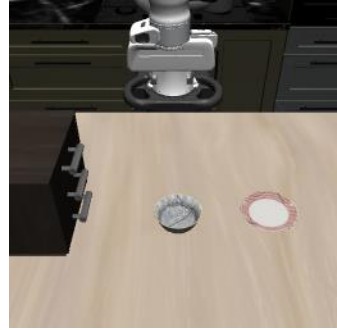

Task: close the top
drawer of the cabinet

**Task: open the top
drawer of the cabinet**

90% success rate
OmniActor-EA  ⟹  90% success rate
OmniActor

**40%** success rate
OmniActor-EA  ⟹  **70%** success rate
OmniActor

Figure 11: The case study about the transfer from GUI to embodied tasks. GUI data enhance the model ability to understand key terms such as "top" or "bottom" in the instructions of embodied tasks, thereby helping the model distinguish similar objects at different positions.

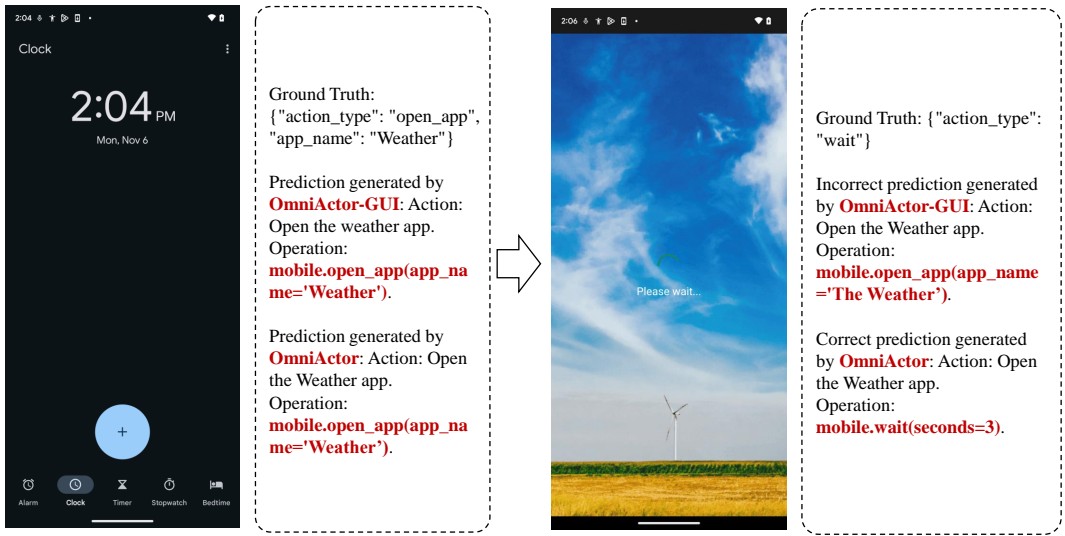

Figure 12: The case study about the transfer from embodied to GUI tasks. Embodied data enhance the model ability to recognize detailed visual cues for GUI tasks.

of the cabinet", where the target region ("open drawer") has significant features, so the success rates of both OmniActor-EA and OmniActor exceed 90%. The other task is "open the top drawer of the cabinet", where the model needs to identify the target drawer among three similar drawers. Thus, the success rate of OmniActor-EA is only 40%, while that of OmniActor reaches 70%.

**The transfer from embodied to GUI tasks**: Embodied data enhance the model ability to recognize detailed visual cues for GUI tasks. We observe a phenomenon: without embodied data, the model may repeatedly output similar GUI actions. We suspect this is because the model fails to effectively capture detailed visual cues to generate new actions. As shown in Figure 12, the screen has changed; however, OmniActor-GUI still outputs the action `open_app`. This may be because OmniActor-GUI fails to capture the visual cue "Please wait..." on the screen. In contrast, OmniActor can correctly output the action `moble.wait`.

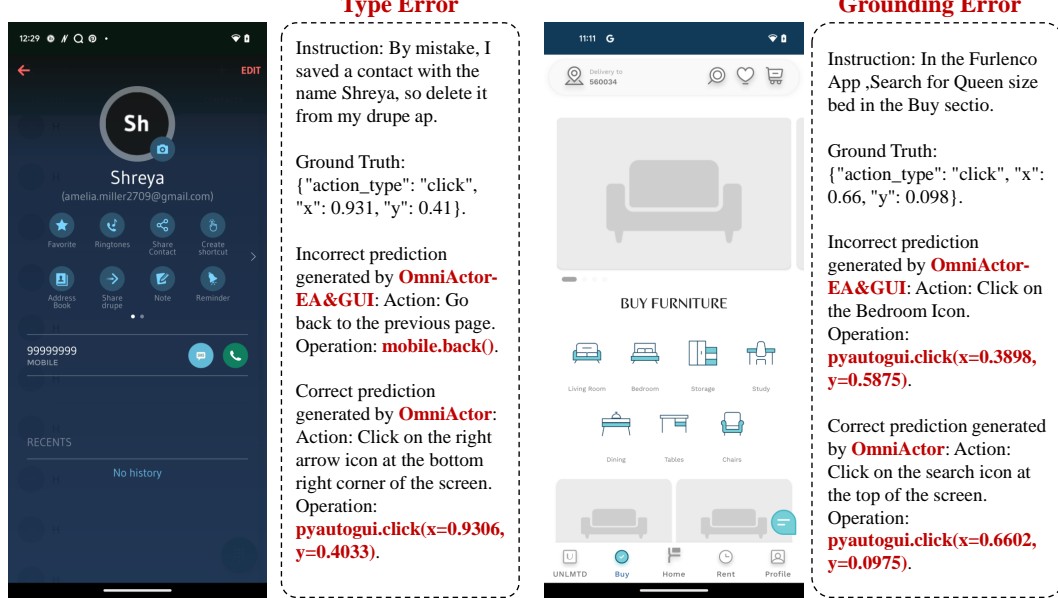

Figure 13: The case study about the performance degeneration brought from data conflict. We present two examples representing two typical types of errors in GUI tasks.

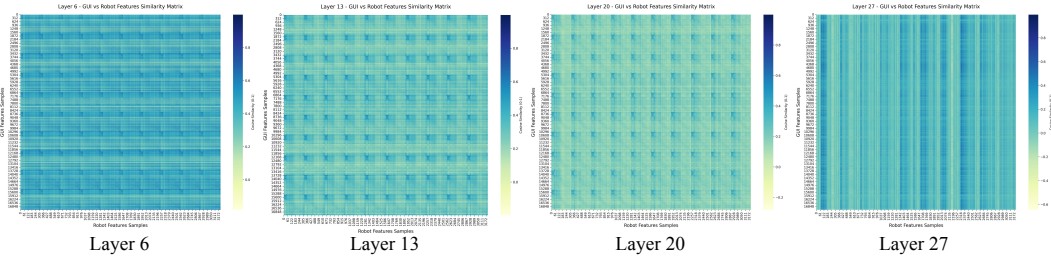

| Layer 6 | Layer 13 | Layer 20 | Layer 27 |

Figure 14: Visualization about synergy and conflict. We analyze the similarity relationships between GUI and embodied features in different layers. Darker colors indicate higher feature similarity, while lighter colors indicate lower feature similarity.

**OmniActor-EA&GUI** *vs.* **OmniActor**: Compared with OmniActor, OmniActor-EA&GUI performs worse on almost all tasks due to the severe conflict between GUI and embodied data. As shown in Figure 13, we present two examples representing two typical types of errors in GUI tasks: in the first example, OmniActor-EA&GUI incorrectly predicts the action type (a type error), while OmniActor makes a correct prediction; in the second example, OmniActor-EA&GUI predicts the correct action type but fails to identify the accurate click position (a grounding error), whereas OmniActor achieves correct prediction.

## 6.7 VISUALIZATION STUDY

We conduct a more in-depth analysis of shallow and deep features. Specifically, we randomly select a subset of GUI data and use OmniActor-GUI to extract intermediate-layer features $f_{\text{gui}} \in R^{N_{\text{gui}} \times C}$, where $N_{\text{gui}}$ indicates GUI sample count. Similarly, we randomly select a subset of embodied data and use OmniActor-EA to extract intermediate-layer features $f_{\text{robot}} \in R^{N_{\text{robot}} \times C}$, where $N_{\text{robot}}$ indicates GUI sample count. We then calculate the similarity matrix between $f_{\text{gui}}$ and $f_{\text{robot}}$. We focus our analysis on the layers $\{6, 13, 20, 27\}$. In other words, we generate four similarity matrices. By visualizing these similarity matrices in Figure 14, we observe that the similarity of shallow features is significantly higher than that of deep features. The mean similarity is 0.4206, 0.3018, 0.2000, and

0.1134 for layer 6, 13, 20, and 27, respectively. This phenomenon further indicates that GUI and embodied tasks exhibit synergy at the shallow layers while showing conflict at the deep layers.

## 6.8 DETAILS ABOUT THE USE OF LARGE LANGUAGE MODELS

We use Large Language Models (LLMs) to aid or polish writing. In detail, we firstly write all the content to complete the initial draft. After finishing the draft, we use a large language model to polish the sentences, mainly for the Methodology and Experiments sections, *e.g.*, "Training data" in Section 4.1. The used prompt is "Please polish the following text and return LaTeX-style text:".

