# OpenReview forum: "OmniActor: A Generalist GUI and Embodied Agent for 2D&3D Worlds"
_ICLR.cc/2026/Conference — ICLR 2026 Poster_

### Official Review · Reviewer_YycE · 2025-10-31

**Soundness:** 3
**Presentation:** 3
**Contribution:** 3
**Rating:** 6
**Confidence:** 4

**Summary:**

This paper explores the synergy between training generalist agents on UI and embodied action data. OmniActor is a generalist agent across embodied and UI tasks, leveraging a mixture of expert architecture for deep layers to separate the UI and embodied knowledge. Empirical results demonstrate OmniActor outperforming training with only one data source and prior UI or embodiment-specific baselines.

**Strengths:**

1. The paper addresses the important problem of generalist models between UI and embodiment data. Both domains already have limited amounts of interactive data. Therefore, it's important to leverage the synergies from both to improve performance.

1. The results in Table 1 show that OmniActor outperforms baselines in UI control tasks and closely matches specialized embodiment-specific baselines in embodied tasks.

1. Through the proposed parameter update similarity metric, the paper illustrates how the parameter update diverges for deeper layers between the two data modalities in Figure 4. This motivates the intuitive solution of an MoE for deeper layers.

**Weaknesses:**

1. The paper does not sufficiently explore the ways of disentangling the UI and embodiment knowledge, and only explores one MoE approach. For example, is it possible to share only the FFN or attention modules as well, or is it necessary to separate both? This is a key contribution of the work and thus should be sufficiently analyzed.

1. The paper does not analyze how to select the "deep layer threshold" K value (L468). Likewise, the paper does not analyze how robust the training is to the choice of K. This value is a key hyperparameter that could have a large impact on performance since it addresses the core problem of conflicting optimization directions. It is not clear how to select K based on the visualization in Figure 4 alone. The parameter update similarity appears similar for deeper layers as well in Figures 4a and 4b.

1. The paper does not compare to other VLA models in embodied tasks, such as OpenVLA or pi0. These VLA models are more directly comparable to OmniActor since they are also based on a VLM, are more recent, and perform better on the Libero benchmark.

1. For embodied tasks, the paper only evaluates on the Libero benchmark. Likewise, only data from Libero is used for training. Only evaluating on a single embodied benchmark is contrary to the claim that OmniActor is a "generalist agent". A generalist agent should be evaluated on multiple embodiment benchmarks (for example, CALVIN or SimplerEnv).

1. OmniActor achieves strong performance relative to baselines largely because of the starting OmniActor framework rather than a strong cross-domain transfer. The OmniActor, only trained on single-domain data in GUI or embodied tasks, already outperforms most baselines in Table 1.

1. The paper does not analyze which capabilities are transferring between GUI and embodied domains, just that there is broad transfer happening on the benchmarks. Which tasks in Libero and the GUI tasks is the embodiment data improving performance in? This is important for understanding why this cross-domain data is helpful. Likewise, it is important to study for which individual subtasks or behaviors the cross-domain data hurts performance in with the default training to better understand where OmniActor helps.

**Questions:**

1. How does performance vary with different choices of K?

1. How does OmniActor handle varying resolutions between GUI and embodiment data? Other GUI benchmarks, like AndroidWorld, have high resolution images (2400x1080) which are necessary to perceive and interact with all the GUI elements. On the other hand, Libero images are 256x256. L676 says all the images are resized to 448x448, but how would the OmniActor framework train with high-resolution GUI images as well?

1. Why is does OmniActor trained with only the GUI data achieve such higher performance than prior works across generalist and GUI agents?

1. Why is the transfer from GUI to embodied tasks higher than vice versa? Is it because there is more GUI data?

1. Why use distinct prediction heads for GUI and embodied actions, since the vocabulary of these action spaces is distinct? Since the action space is distinct, the token embeddings do not overlap, so there is no difference in splitting versus keeping the embeddings the same.

---

> ### Author Response · Authors · 2025-11-27
> **Response to Reviewer YycE (Part 1)**
>
> We sincerely thank reviewer **YycE** for the thoughtful and constructive feedback, which will help improve our work. Our responses are as follows.
>
> > *W1: Is it possible to share only the FFN or attention modules as well, or is it necessary to separate both*?
>
> **Response**: We explore different ways of disentangling the GUI and embodied knowledge in the table below. OmniActor separate the parameters of attention, FFNs, and prediction heads. OmniActor-attention shares attention while seperates the remaining parameters. OmniActor-FFN shares FFNs and OmniActor-head shares the prediction head. We find that sharing attention shows the least performance degradation, while sharing FFNs orheads leads to significant performance drops. This may be attributed to that FFNs account for a large proportion of the total parameters, and heads are crucial for prediction.
>
> |                     | LIBERO-90 | AndroidControl-Low | AndroidControl-High | GUI Odyssey | Avg  |
> | ------------------- | --------- | ------------------ | ------------------- | ----------- | ---- |
> | OmniActor           | 69.5      | 87.5               | 77.1                | 66.0        | 75.0 |
> | OmniActor-attention | 68.0      | 86.5               | 76.7                | 63.7        | 73.7 |
> | OmniActor-FFN       | 66.0      | 84.6               | 74.0                | 63.2        | 72.0 |
> | OmniActor-head      | 61.0      | 86.0               | 75.7                | 62.6        | 71.3 |
>
> > *W2 and Q1: How robust the training is to the choice of $K$.*
>
> **Response**: $K$ is a tunable hyper-parameter. We discuss different selections of $K$ in the following table and find that the performance is optimal when $K$=8. In addition, we find that we previously used both $K$ and $T$ to denote the layer depth threshold, and we have now standardized it to $T$ in the revised manuscript.
>
> |      | LIBERO-90 | AndroidControl-Low | AndroidControl-High | GUI Odyssey | Avg  |
> | ---- | --------- | ------------------ | ------------------- | ----------- | ---- |
> | $K$=4  | 63.0      | 84.5               | 74.1                | 65.4        | 71.8 |
> | $K$=8  | 69.5      | 87.5               | 77.1                | 66.0        | 75.0 |
> | $K$=12 | 63.0      | 87.7               | 75.8                | 67.7        | 73.6 |
> | $K$=20 | 61.5      | 87.3               | 71.7                | 67.6        | 72.0 |
>
> > *W3: The paper does not compare to other VLA models in embodied tasks, such as OpenVLA or $\pi_0$.*
>
> **Response**: We supplement SpatialVLA [1], OpenVLA [2], and $\pi_0$ [3] in the table below. It is noted that compared with OmniActor, OpenVLA and $\pi_0$ use significantly more embodied data. For example, $\pi_0$ uses a large amount of in-house embodied data, while OmniActor only uses LIBERO as embodied training data. The current data construction of OmniActor mainly focuses on GUI tasks. We plan to continue to expand the embodied data in the future. We have incorporated these works in **Table 1** of the revised manuscript.
> | | SpatialVLA | OpenVLA | $\pi_0$ | OmniActor |
> | --------- | ---------- | ------- | ---- | --------- |
> | LIBERO-90 | 46.2       | 73.5    | 87.3 | 69.5      |
>
> [1] SpatialVLA: Exploring Spatial Representations for Visual-Language-Action Models, RSS 2025.
>
> [2] OpenVLA: An Open-Source Vision-Language-Action Model, CORL 2024.
>
> [3] $\pi_0$: A Vision-Language-Action Flow Model for General Robot Control, arxiv 2024.
>
> > *W4: A generalist agent should be evaluated on multiple embodiment benchmarks (for example, CALVIN or SimplerEnv).*
>
> **Response**: We conduct experiments by replacing LIBERO with real-world robot data, and the experimental results show the superiority of OmniActor under real-world embodied benchmark. More details please refer to **Section 6.3** of the revised manuscript.  We are still preparing more experimental results of this part, and we will report them soon.
>
> > *W5 and Q3: Why is does OmniActor trained with only the GUI data achieve such higher performance than prior works across generalist and GUI agents?*
>
> **Response**: We have invested significant efforts in GUI data to collect high-quality and large-scale GUI data for training models. Thus, OmniActor-GUI outperforms most baseline models in GUI tasks. To promote the development of the agent community, we will release all the code and data.

---

> > ### Public Comment · ~Matteo_Jiahao_Chen1 · 2026-02-25
> > **Code, dataset and model weights release**
> >
> > Hi, thanks for the interesting work!
> >
> > I was wondering whether the code and the GUI dataset mentioned in the paper are already available somewhere (e.g., GitHub or Hugging Face), or if there is a planned release link.
> >
> > Also, do you plan to release the trained model weights as well?
> >
> > Thanks!

---

> ### Author Response · Authors · 2025-11-27
> **Response to Reviewer YycE (Part 2)**
>
> > *W6: Which tasks in Libero and the GUI tasks is the embodiment data improving performance in?*
>
> **Response**: We further analyze the synergy and conflict between GUI and embodied tasks.
>
> **The transfer from GUI to embodied tasks**: GUI data enhance the model ability to understand key terms such as ''top" in the instructions of embodied tasks, so helping the model distinguish similar objects at different positions. We present two tasks in **Figure 11** of the revised manuscript: one task is "close the top drawer of the cabinet", where the target region ("open drawer") has significant features, so the success rates of both OmniActor-EA and OmniActor exceed 90%. The other task is "open the top drawer of the cabinet", where the model needs to identify the target drawer among three similar drawers. Thus, the success rate of OmniActor-EA is only 40%, while that of OmniActor reaches 70%.
>
> **The transfer from embodied to GUI tasks**: Embodied data enhance the model ability to recognize detailed visual cues for GUI tasks. We observe a phenomenon: without embodied data, the model may repeatedly output similar GUI actions. We suspect this is because the model fails to effectively capture detailed visual cues to generate new actions. As shown in **Figure 12** of the revised manuscript, the screen has changed; however, OmniActor-GUI still outputs the action open_app. This may be because OmniActor-GUI fails to capture the visual cue "Please wait..." on the screen. In contrast, OmniActor can correctly output the action moble.wait.
>
> **OmniActor-EA&GUI vs. OmniActor**: Compared with OmniActor, OmniActor-EA&GUI performs worse on almost all tasks due to the severe conflict between GUI and embodied data. As shown in **Figure 13** of the revised manuscript, we present two examples representing two typical types of errors (Type Error and Grounding Error) in GUI tasks.
>
> More details please refer to **Section 6.6** of the revised manuscript.
>
> > *Q2: How would the OmniActor framework train with high-resolution GUI images as well?*
>
> **Response**: We use the normal training recipe. Specifically, all images are processed into tokens by the visual encoder of Qwen2/2.5-VL by dynamic resolution technique. Higher resolution corresponds to more tokens while lower resolution results in fewer tokens. Thus, the level of image resolution has no impact on model training; it only affects the number of tokens processed by the model.
>
> > *Q4: Why is the transfer from GUI to embodied tasks higher than vice versa?*
>
> **Response**: Based on our case study, we hypothesize the following: *GUI data enhances the model ability to understand key terms in the instructions of embodied tasks, while embodied data enhance the model ability to recognize detailed visual cues for GUI tasks*. Among these two types of transfer, the latter may only exist in certain hard cases of GUI tasks, but the former, instruction comprehension, is crucial for almost all embodied tasks. Thus, the transfer from GUI to embodied tasks is higher than vice versa.
>
> > *Q5: Why use distinct prediction heads for GUI and embodied actions, since the vocabulary of these action spaces is distinct?*
>
> **Response**: The reason is consistent with that behind "sharing shallow layers and separating deep layers". We find that the deeper the layers, the more severe the gradient conflicts of data in parameters, with the prediction head being the most affected. Thus, it is necessary to separate the head. We add a supplementary data point about *parameter update similarity* of the head in **Figure 10** of the revised manuscript. In addition, **the study about OmniActor-head (response to _W1_)** also verifies the importance of seperating the head.

---

### Official Review · Reviewer_h7cP · 2025-11-01

**Soundness:** 3
**Presentation:** 2
**Contribution:** 3
**Rating:** 4
**Confidence:** 3

**Summary:**

This paper studies unified 2D GUI and 3D embodied control with a single MLLM, showing naïve joint training hurts because optimization is synergistic in shallow layers but conflicting in deeper ones. It introduces a layer-heterogeneous MoE that shares shallow blocks but splits deep experts with task-type routing to reconcile domains. It yields consistent gains compared with baselines.

**Strengths:**

* The paper is well-motivated, and the proposed method makes sense.

* It shows improvements over both training separate single-domain agents and naïvely mixing all data into one model.

**Weaknesses:**

* The gains on benchmarks are modest and may fall within variance (e.g., 77.1 vs. 74.5 on GUI tasks).

* In section 4.3, the authors propose a metric called "parameter update similarity". Although they show differences in GUI and Embodied groups, more evidence about in-group metric is needed to further support the claim. If in-group metric shows similar trend then the claim may be incorrect.

* Table 2 should include a MoE variant without explicit expert separation to test whether improvements stem from MoE capacity rather than task-specific partitioning.


* The appendix should provide fuller experimental details (prompts, representative training samples, and evaluation samples) to support reproducibility.

* Another limitation of this work is that the scope is quite limited with only offline GUI control and one simulated robot task. Including online GUI evaluation and real-robot experiments (e.g., as in Magma) would broaden the contribution.

**Questions:**

Please refer to the weakness part.

---

> ### Author Response · Authors · 2025-11-27
> **Response to Reviewer h7cP (Part 1)**
>
> We sincerely thank reviewer **h7cP** for the thoughtful and constructive feedback, which will help improve our work. Our responses are as follows.
>
> > *W1: The gains on benchmarks are modest and may fall within variance.*
>
> **Response**: We conduct five independent training runs for OmniActor-GUI and OmniActor respectively, and calculate their statistical mean and standard deviation. As shown in the table below, the performance gain is statistically reliable.
>
> |               | AndroidControl-Low | AndroidControl-High | GUI Odyssey  |
> | ------------- | ------------------ | ------------------- | ------------ |
> | OmniActor-GUI | 88.82 ± 1.27       | 75.20 ± 0.78        | 63.77 ± 0.32 |
> | OmniActor     | 88.98 ± 1.53       | 77.30 ± 0.85        | 65.67 ± 0.31 |
>
> > *W2: More evidence about in-group metric is needed.*
>
> **Response**: We have randomly divided the GUI training data into two in-groups, named GUI-1 and GUI-2 respectively. GUI-1 and GUI-2 are roughly equal in size, each accounting for 50% of the total GUI training data. Then, we train the model using GUI-1 and GUI-2 separately. Based on the trained models, the *parameter update similarity* is computed. The results are presented in **Figure 9** of the revised manuscript. We find that the *parameter update similarity* across different layers is consistently high, and there is no trend of "higher similarity in shallow layers and lower similarity in deep layers". This further verifies the necessity of designing layer-heterogeneity MoE for GUI and embodied data. More details please refer to **Section 6.5** of the revised manuscript.
>
> > *W3: Table 2 should include a MoE variant without explicit expert separation.*
>
> **Response**: We design the OmniActor *router*: a router is incorporated prior to each MoE layer, which takes token features as input and outputs the probabilities of assigning the token to different experts. The expert with the highest probability is selected as the active expert for the token. The expert number is set to 2, which is consistent with that in OmniActor. The following table shows the superiority of OmniActor over OmniActor *router*, which verifies that improvements stem from task-specific partitioning rather than MoE capacity.
>
> |                    | LIBERO-90 | AndroidControl-Low | AndroidControl-High | GUI Odyssey | Avg  |
> | ------------------ | --------- | ------------------ | ------------------- | ----------- | ---- |
> | OmniActor          | 69.5      | 87.5               | 77.1                | 66.0        | 75.0 |
> | OmniActor *router* | 64.0      | 86.0               | 72.6                | 66.1        | 72.2 |
>
> > *W4: Fuller experimental details (prompts, representative training samples, and evaluation samples) to support reproducibility.*
>
> **Response**: We have provided fuller experimental details in **Section 6.2** of the revised manuscript,  inlcluding the prompts we use in this work and representative GUI and embodied samples. Furthermore, we have included detailed case studies in **Section 6.6**.

---

> ### Author Response · Authors · 2025-11-27
> **Response to Reviewer h7cP (Part 2)**
>
> > *W5: Real-robot experiments and online GUI evaluation.*
>
> **Response**: We use the real-robot data to conduct experiments. Specifically, we collect some real-world robot data. Then, we convert the format of these data into the embodied data format described in the main text, which is used to replace the LIBERO dataset for training both OmniActor-EA and OmniActor. Subsequently, we select four unseen tasks to evaluate the performance of the trained models: "move redbull can near sponge", "open top drawer", "place green jalapeno chip bag into top drawer", "pick orange". We perform 20 trials per task. We present an example of the real-world robot data in **Figure 7** of the revised manuscript, illustrating the motion of the robotic arm in a 3D real-world environment. Meanwhile, the performance is reported in **Figure 8** of the revised manuscript. We also provide data points (task sucess rate) in the below table. The experimental results demonstrate that OmniActor achieves a success rate exceeding 60%, and its performance outperforms OmniActor-EA, thereby validating the generalization of the Layer-heterogeneous MoE under 3D real-world robot data. More details please refer to **Section 6.3** of the revised manuscript.
>
>
> |              | move redbull can near sponge | open top drawer | place green jalapeno chip bag into top drawer | pick orange | Avg |
> | ------------ | ---------------------------- | --------------- | --------------------------------------------- | ----------- | --- |
> | OmniActor-EA | 45%                          | 35%             | 65%                                           | 65%         | 53% |
> | OmniActor    | 55%                          | 55%             | 80%                                           | 70%         | 65% |
>
> Besides, this paper focuses on analyzing conflicts and synergies between GUI and embodied data, and proposes the Layer-heterogeneous MoE to alleviate conflicts and leverage synergies. This core mechanism has been fully validated in offline scenarios. Online GUI evaluation requires the construction of a dedicated real-time interaction framework, involving significant engineering implementation costs. It is difficult to implement in the short term and falls beyond the core scope of this work. We have initiated the development of an interactive environment for online GUI evaluation, and plan to complete this work before we open-source the code in the future.

---

### Official Review · Reviewer_HVrX · 2025-11-01

**Soundness:** 3
**Presentation:** 3
**Contribution:** 2
**Rating:** 4
**Confidence:** 3

**Summary:**

This paper introduces a unified multimodal agent that can perform both 2D GUI and 3D embodied tasks. The authors find that naive joint training causes conflicts between modalities, so they propose a Layer-Heterogeneity Mixture-of-Experts architecture that shares shallow layers and separates deep layers. Trained on large-scale standardized datasets, OmniActor achieves state-of-the-art or superior performance across both GUI and embodied benchmarks, demonstrating effective generalization between virtual and physical environments.

**Strengths:**

1. This paper presents a clear insight into the conflict–synergy relationship between GUI and embodied data, supported by empirical analysis.
2. The proposed Layer-Heterogeneity MoE is biologically inspired, effectively balancing parameter sharing and specialization.
3. The authors demonstrate strong experimental results, outperforming both domain-specific and prior generalist models across multiple benchmarks.

**Weaknesses:**

1. The text on the left side of Figure 3 is too small to read clearly.
2. Please verify whether LIBERO-90 should be the training dataset. Typically, LIBERO-10 is used as the test set.
3. Table 1 could be improved by including some recent works on LIBERO published in 2025.
4. I suggest that the authors include real-robot experiments to demonstrate the algorithm’s generalization capability in the 3D physical world.
I5. It would be valuable to add backbone ablation studies, such as experiments using Qwen-3.

**Questions:**

1. Why did the authors choose to fuse 2D and 3D scenes specifically using robot manipulation and GUI grounding tasks? Does this choice lead to excessive data heterogeneity?
2. Could this design cause a cross-domain data imbalance issue, and if so, how is it addressed?
3. In the dataset, are the GUI and robot data quantities balanced? Should more diverse robot datasets be incorporated?

---

> ### Author Response · Authors · 2025-11-27
> **Response to Reviewer HVrX (Part 1)**
>
> We sincerely thank reviewer **HVrX** for the thoughtful and constructive feedback, which will help improve our work. Our responses are as follows.
>
> > *W1: The text on the left side of Figure 3 is too small to read clearly.*
>
> **Response**: We have enlarged the text size on the left side of Figure 3. Please refer to **Figure 3** in the revised manuscript for details.
>
> > *W2: Please verify whether LIBERO-90 should be the training dataset.*
>
> **Response**: LIBERO-90 training data is introduced in [1]. We follow some works, *e.g.*, [2,3], to use LIBERO-90 training data and LIBERO-90 test benchmark to conduct experiments. One reason that we use LIBERO-90 is that LIBERO-90 has a larger dataset size than LIBERO-10 and contains 90 robot tasks, which is sufficiently diverse.
>
> [1] LIBERO: Benchmarking Knowledge Transfer for Lifelong Robot Learning, arxiv 2023.
>
> [2] PRISE: LLM-Style Sequence Compression for Learning Temporal Action Abstractions in Control, ICML 2024.
>
> [3] Multimodal Diffusion Transformer: Learning Versatile Behavior from Multimodal Goals, RSS 2024.
>
> > *W3: Some recent works on LIBERO published in 2025.*
>
> **Response**: As shown in the table below, we have included some recent works [4,5] on LIBERO published in 2025. It is noted that compared to OmniActor, UniACT uses significantly more embodied data, unifying Open-X Embodiment, LIBERO, and Droid for training, while OmniActor only uses LIBERO as embodied training data. The current data construction of OmniActor mainly focuses on GUI tasks. We plan to continue to expand the embodied data in the future. We have incorporated these works in **Table 1** of the revised manuscript.
>
> | Methods   | SpatialVLA | UniACT | OmniActor |
> | --------- | ---------- | ------ | --------- |
> | LIBERO-90 | 46.2       | 73.0   | 69.5      |
>
> [4] SpatialVLA: Exploring Spatial Representations for Visual-Language-Action Models, RSS 2025.
>
> [5] Universal Actions for Enhanced Embodied Foundation Models, CVPR 2025.
>
>
> > *W4: real-robot experiments and backbone ablation studies.*
>
> **Response**: We use the real-robot data to conduct experiments. Specifically, we collect some real-world robot data. Then, we convert the format of these data into the embodied data format described in the main text, which is used to replace the LIBERO dataset for training both OmniActor-EA and OmniActor. Subsequently, we select four unseen tasks to evaluate the performance of the trained models: "move redbull can near sponge", "open top drawer", "place green jalapeno chip bag into top drawer", "pick orange". We perform 20 trials per task. We present an example of the real-world robot data in **Figure 7** of the revised manuscript, illustrating the motion of the robotic arm in a 3D real-world environment. Meanwhile, the performance is reported in **Figure 8** of the revised manuscript. We also provide data points (task success rate) in the below table. The experimental results demonstrate that OmniActor achieves a success rate exceeding 60%, and its performance outperforms OmniActor-EA, thereby validating the generalization of the Layer-heterogeneous MoE under 3D real-world robot data. More details please refer to **Section 6.3** of the revised manuscript. Regarding the backbone ablation studies, we have provided related discussion in **Table 3** of the manuscript. The experimental results verify the generalization of OmniActor across different base MLLMs (the most widely used Qwen2-VL and Qwen2.5-VL).
>
>
> |              | move redbull can near sponge | open top drawer | place green jalapeno chip bag into top drawer | pick orange | Avg |
> | ------------ | ---------------------------- | --------------- | --------------------------------------------- | ----------- | --- |
> | OmniActor-EA | 45%                          | 35%             | 65%                                           | 65%         | 53% |
> | OmniActor    | 55%                          | 55%             | 80%                                           | 70%         | 65%

---

> ### Author Response · Authors · 2025-11-27
> **Response to Reviewer HVrX (Part 2)**
>
> > *Q1: Why did the authors choose to fuse 2D and 3D scenes specifically using robot manipulation and GUI grounding tasks? Does this choice lead to excessive data heterogeneity?*
>
> **Response**: Robot manipulation and GUI grounding tasks are key tasks for evaluating the ability of an agent to interact with 3D and 2D worlds, respectively. Humans usually need to alternately execute embodied and GUI actions when handling complex real-world tasks, as discussed in [6]. For example, when a person is instructed to "Cook apple bread with fried eggs, potatoes, and lettuce using a pan", he/she may first search for a recipe online (a GUI task). Suppose the first step of the recipe is to "crack an egg". The person needs to pick up an egg and crack it into the pan (an embodied task). Next, he/she needs to check the next step of the recipe (a GUI task) and then execute the subsequent embodied action. When he/she finds that a certain ingredient is missing, he/she can place an order online (a GUI task). Therefore, we use robot manipulation and GUI grounding tasks to fuse 3D and 2D scenes, with the aim of enabling agents to handle these complex real-world tasks. Then, expanding the interaction environments of the agent will inevitably enlarge data heterogeneity, regardless of the specific tasks for such expansion. Thus, the conflict-synergy phenomenon is general and worthy of discussion, and the Layer-heterogeneous MoE proposed in this paper effectively eliminates conflicts while leveraging synergies.
>
> [6] Embodied Web Agents: Bridging Physical-Digital Realms for Integrated Agent Intelligence, arxiv 2025.
>
> > *Q2 and Q3: Could this design cause a cross-domain data imbalance issue, and if so, how is it addressed? In the dataset, are the GUI and robot data quantities balanced? Should more diverse robot datasets be incorporated?*
>
> **Response**: Our GUI and robot data contain 706K and 669K samples respectively, with a ratio close to 1:1. This is one reason that we selected LIBERO-90 as the training set instead of LIBERO-10; LIBERO-90 has a larger dataset size than LIBERO-10 and contains 90 robot tasks, which is sufficiently diverse. Furthermore, more diverse robot datasets should be  incorporated when the agent needs to handle more complex embodied tasks. Intuitively, as more diverse robot datasets are incorporated, the conflict-synergy problem among data will become more significant and the proposed method can achieve a higher performance increase.

---

### Official Review · Reviewer_kuDc · 2025-11-04

**Soundness:** 4
**Presentation:** 4
**Contribution:** 4
**Rating:** 10
**Confidence:** 5

**Summary:**

This paper introduces OmniActor: a unified vision-language-action agent for GUI and Robotic applications.

There are two main contributions in this paper:

1- Solving the common data conflict problem for generalist agents by observing that the data conflict leads to gradient updates that are consistent in shallow layers, but divergent at deeper layer. The solution suggested to this is by sharing the parameters in the shallow layers but specializing the experts (weights) for the deeper layer closer to the action decision.

2- Unifying the data and action formats across both GUI and Robotics domains.

The paper has multiple evaluations on various GUI and Robotic benchmarks, and using multiple base models, showing significant uplift and the ability to generalize.

**Strengths:**

- The paper makes a very insightful observation about the data conflict problem, which is an important challenge for generalist agents.

- The change suggested to solve the data conflict problem is rather simple and easy to implement, and the results show significant improvement.

- The paper is very well written and illustrated.

- The ablation studies and changing the base model evals are very well done. Makes me trust the results more.

**Weaknesses:**

- The suggested separation method seems arbitrary in a few design decisions (e.g. how to decide on the T parameter for shared layers properly, and the granularity of the separation/specialization: why separate GUI and Robotic only but not Mobile and Computer Use)

- I have doubts about how this separation/specialization method would scale if we have lots of domains not just two.

- The paper does not use any kind of RL to improve the reasoning like other similar systems. Would be interesting to see how RL can be combined with this finding to produce even better numbers.

**Questions:**

- The separation/specialization algorithm used here is manual and depends heavily on similarity statistics (i.e. check the instance type gui/robotics, then route to the correct set of weights). Do you think that this can be automated within the neural architecture itself (i.e. make it a part of the learning process)?

---

> ### Author Response · Authors · 2025-11-22
> **Response to Reviewer kuDc**
>
> We sincerely thank reviewer **kuDc** for the thoughtful and constructive feedback, which will help improve our work. Our responses are as follows.
>
> > *W1: How to decide on the $T$ parameter for shared layers properly, and the granularity of the separation/specialization.*
>
> **Response**: The layer depth threshold $T$ is a tunable hyper-parameter. We discuss different selections of $T$ in the following table and find that the performance is optimal when $T$=8. Regarding the granularity of separation, we think that each environment can be assigned a corresponding expert. The paper sets two experts mainly to verify the effectiveness of layer-heterogeneity MoE at the minimum cost. Theoretically, a finer-grained division is fully feasible and should yield performance improvements.
>
> |      | LIBERO-90 | AndroidControl-Low | AndroidControl-High | GUI Odyssey | Avg  |
> | ---- | --------- | ------------------ | ------------------- | ----------- | ---- |
> | $T$=4  | 63.0      | 84.5               | 74.1                | 65.4        | 71.8 |
> | $T$=8  | 69.5      | 87.5               | 77.1                | 66.0        | 75.0 |
> | $T$=12 | 63.0      | 87.7               | 75.8                | 67.7        | 73.6 |
> | $T$=20 | 61.5      | 87.3               | 71.7                | 67.6        | 72.0 |
>
> > *W2: How this separation/specialization method would scale if we have lots of domains not just two.*
>
> **Response**: Scaling to more domains can be addressed by scaling the number of experts, and action space differences may be the key factor for domain separation. In detail,  we conduct an analysis of *parameter update similarity* in **Figure 4**, and have supplemented in-group *parameter update similarity* and *parameter update similarity* of the head in **Figures 9** and **Figure 10** of the revised manuscript. Combining these analyses, we find that action space differences should be important; when there are significant action differences across multiple domains, allocating one expert to each domain and adopting the Layer-heterogeneous MoE strategy should yield satisfactory results.
>
> > *W3: How RL can be combined with this finding to produce even better numbers.*
>
> **Response**: We focus on analyzing conflicts and synergies between GUI and embodied data, and propose the Layer-heterogeneous MoE to alleviate conflicts and leverage synergies. This core mechanism has been fully validatedd in the SFT phase. Given the thorough validation in SFT, we have initiated RL experiments, serving as a further optimization for agents. However, RL requires the construction of a dedicated real-time interaction framework, involving significant engineering implementation costs. It is difficult to implement in the short term. We plan to continue the RL training to develop a new model, possibly naming OmniActor-v2, and commit to completing this work when we open-source the code in the future.
>
> > *Q1: Do you think that this can be automated within the neural architecture itself (i.e. make it a part of the learning process)?*
>
> **Response**: This is a highly insightful question. Our idea is the multi-agent framework: we can construct a workflow where the Planner determines the separation scheme, modifies the base MLLMs to generates the Actor according to this scheme. Specifically, we sample training data from different environments and feed the sampled data to the Planner. The Planner first extracts the gradients of these data across different layers of the base MLLMs, then predicts the separation scheme based on the gradients; this scheme may include separation layer IDs, separation granularity, and so on. Based on the separation scheme, the Planner invokes code tools to modify the model to generate the Actor. Finally, we can jointly optimize the Planner and the Actor via RL. This could become a highly promising future research direction. We are also already preparing for the training of OmniActor-v2 and will incorporate this into the training plan. When we open-source the code in the future, we should be able to complete the initial design and validation.

---

### Author Response · Authors · 2025-11-27
**List of Revisions**

We sincerely appreciate your thoughtful and constructive feedback. We attach great importance to every comment and suggestion. According to the reviewers' suggestions, we have submitted a revised version of the paper. The main revisions are as follows:

-In **Table 1**, we have added **the recent embodied works for comparison**, to address the concerns of insufficient comparison raised by HVrX and YycE.

-In **Table 2**, we have included **a MoE variant without explicit expert separation**, to address the concerns of the effectiveness of task-specific partitioning raised by reviewer h7cP.

-In **Section 6.2**, we have provided **fuller experimental details**, inlcluding the prompts we use in this work and representative GUI and embodied samples, to address the concerns of reproducibility raised by reviewer h7cP.

-In **Section 6.3**, we have provided **real-robot experiments**, to address the concerns of genenralization in the 3D physical world raised by reviewers HVrX and h7cP.

-In **Section 6.4**, we have provided **the ablation study about the layer depth threshold $T$**, to address the concerns of the selection of $T$ raised by reviewers kuDc and YycE.

-In **Section 6.4**, we have provided **the statistical mean and standard deviation of the model performance**, to address the concerns of performance variance by reviewer h7cP.

-In **Section 6.4**, we have provided **the ablation study about different disentangling way**s, to address the concerns of disentangling way design raised by reviewer YycE.

-In **Section 6.5**, we have provided **the discuss about the in-group *parameter update similarity* and the *parameter update similarity* on the head**, to address the concerns of GUI and embodied difference raised by reviewers h7cP and the concerns of head separation design raised by YycE.

-In **Section 6.6**, we have provided **detailed case study**, to address the concerns of lacking the detailed analysis for transfer behaviors raised by reviewers YycE.

-In addition, we have modified Figure 3, add data description in 4.1, and correct some typo, to address the concerns raised by reviewer HVrX.

---

### Author Response · Authors · 2025-12-03
**Summary of the Discussion Phase and Paper Revisions**

Dear Area Chair,

We sincerely thank you and the reviewers for the time and effort devoted to assessing our submission. We value the constructive feedback received, which has significantly helped us improve the quality of our paper. Below, we summarize the consensus on the paper's strengths and the corresponding revisions made in response to the reviews.

First, we appreciate that all reviewers acknowledge the contribution and novelty of this work:
*   **Insightful motivation:** Reviewers praise the paper for identifying the "important challenge" of data conflict and the synergy between GUI and embodied data (kuDc, HVrX, YycE), describing the work as "well-motivated" (h7cP) and "very insightful" (kuDc).
*   **Effective & intuitive method:** The proposed Layer-Heterogeneity MoE is highlighted as "biologically inspired" and effective in balancing parameter sharing and specialization (HVrX). Reviewers appreciate that the solution is "simple and easy to implement" (kuDc) and supported by "intuitive" metric (YycE).
*   **Strong results & quality:** Reviewers agree that the model demonstrates "strong experimental results", outperforming domain-specific and prior generalist models (HVrX, h7cP, YycE). Additionally, the paper is commended for being "well written and illustrated" with "trustworthy" ablation studies (kuDc).

Second, in response to the weaknesses and questions raised during review and discussion, we have made the following clarifications and additions:

**Revisions to the paper:**

**1. Comparison with related works and baselines (HVrX, YycE, h7cP)**
*   Additional comparison: In **Table 1**, we have added comparisons with recent embodied works (e.g., OpenVLA, pi0) to address concerns regarding insufficient comparison (HVrX, YycE).
*   MoE variant analysis: In **Table 2**, we include an MoE variant without explicit expert separation as a baseline (h7cP), which verifies the effectiveness of the task-specific partitioning.

**2. Generalization and real-world experiments (HVrX, h7cP)**

In **Section 6.3**, we have added real-robot experiments to demonstrate the algorithm’s generalization capability in the 3D physical world, moving beyond simulation.

**3. Reproducibility and experimental details (h7cP, HVrX)**
*   Detailed setup: In **Section 6.2**, we provide fuller experimental details, including the specific prompts used and representative GUI/embodied samples to ensure reproducibility (h7cP).
*   We have optimized the visualization in **Figure 3**, added a detailed data description in **Section 4.1**, and corrected typos to improve clarity (HVrX).

**4. Ablation studies and statistical analysis (kuDc, YycE, h7cP)**
*   Layer depth threshold: In **Section 6.4**, we conduct an ablation study on the layer depth threshold $T$ to justify our hyper-parameter selection (kuDc, YycE).
*   Performance variance: Also in **Section 6.4**, we reporte the statistical mean and standard deviation of the model performance to address concerns about performance variance (h7cP).
*   Disentangling designs: We provide an ablation study exploring different ways of disentangling knowledge in **Section 6.4** (YycE),

**5. Mechanism analysis and case studies (h7cP, YycE)**
*   _Parameter update similarity_: In **Section 6.5**, we discuss the in-group _parameter update similarity_ and the similarity on the head. This addresses concerns regarding the metric trends within groups (h7cP) and the design of head separation (YycE).
*   Case study: In **Section 6.6**, we add detailed case studies to provide a deeper analysis of transfer behaviors and subtask performance (YycE), providing a novel insight "GUI data enhances the model ability to understand key terms in the instructions of embodied tasks, while embodied data enhance the model ability to recognize detailed visual cues for GUI tasks".

**Discussion phase:** We submitted our response and revised manuscript on November 28. Unfortunately, due to the early conclusion of the discussion period, we did not receive any response or further feedback from the reviewers. We have made every effort to incorporate their feedback comprehensively into the revised manuscript. We believe the additional experiments and analyses listed above effectively address the reviewers' concerns regarding comparison, generalization, reproducibility, and model design.

Thank you for your time and consideration. We hope this summary assists in your decision-making process.

Sincerely,

Authors

---

### Meta-Review · Area_Chair_NUge · 2026-01-14

**Summary:**

OmniActor proposes a unified vision-language-action agent for both 2D GUI control and 3D embodied manipulation, arguing that naive joint training creates optimization conflicts that can be mitigated by sharing shallow layers while separating deeper layers via a layer-heterogeneous MoE. The key strengths are a clear empirical motivation (conflict vs. synergy by depth), a simple architecture change, strong multi-benchmark results, and extensive ablations.

The main weaknesses are that some “generalist” claims are still limited by the breadth of embodied evaluation and the reliance on known task-type routing; across reviews, the central concerns were (1) baseline/related-work coverage, (2) robustness/justification of the depth threshold and disentangling choices, (3) variance/reproducibility, and (4) generalization beyond offline settings (e.g., real-robot and/or broader embodied benchmarks). The authors provided extensive experimental results during the rebuttal period to address these concerns.

The submission makes a clear, testable claim about where cross-domain conflicts arise in multimodal agent training. While some “generalist” scope limitations remain (notably broader embodied benchmarks and online GUI/RL evaluation), the revised paper addresses the main review-time technical concerns sufficiently, and the core contribution of layer-heterogeneous sharing vs. specialization as a practical mechanism for resolving GUI/embodiment conflicts appears solid and impactful for the area.

**Reviewer Concerns:**

Several substantive concerns were directly addressed in the rebuttal and revision: the authors added stronger embodied baselines (including recent VLA-style comparisons) and clarified the LIBERO-90 protocol choice. Authors also added statistical mean/std across multiple runs to argue gains exceed variance, an MoE variant intended to disentangle “capacity” vs “task-specific partitioning,” and additional analyses including in-group similarity and head-similarity. Authors also added real-robot experiments (by replacing LIBERO with large-scale real-world robot data and evaluating on held-out tasks), which partially addressed generalization concerns beyond simulation.​

A few concerns remain outstanding or only partially resolved. Online GUI evaluation and RL-style training and some comparisons to agents that leverage interaction/RL remain interesting scenarios to explore. In addition, scalability to many domains (beyond “GUI vs embodied”) and automation of the routing/separation scheme are discussed, but the evidence is mostly qualitative/prospective rather than demonstrated experimentally.​ However, these can be seen as extensions of the idea in this work and are not essential for demonstrating the core contribution.

**Reviewer Scores:**

Reviewers HVrX and h7cP were both at 4 and raised concrete experimental/clarity issues that the revision largely targets: improved Figure 3 readability; clarified LIBERO-90 usage; added more recent comparisons; added real-robot experiments; added variance reporting; added the “no explicit separation” MoE baseline; and expanded reproducibility details. If they had been able to fully engage in discussion, it is plausible that they would have moved from 4 to 6, since their critiques were primarily about missing evidence rather than fundamental flaws.​ Other reviewers support the acceptance of this paper (rating 10 and 6).

---

### Decision · Program_Chairs · 2026-01-26

Accept (Poster)